# Structural elucidation of recombinant *Trichomonas vaginalis* 20S proteasome bound to covalent inhibitors

Jan Silhan [1,7], Pavla Fajtova [1,2,7] ✉, Jitka Bartosova[1], Brianna M. Hurysz[2], Jehad Almaliti[3,4], Yukiko Miyamoto[5], Lars Eckmann[5], William H. Gerwick[4], Anthony J. O'Donoghue [2,6] ✉ & Evzen Boura [1] ✉

The proteasome is a proteolytic enzyme complex essential for protein homeostasis in mammalian cells and protozoan parasites like *Trichomonas vaginalis (Tv)*, the cause of the most common, non-viral sexually transmitted disease. *Tv* and other protozoan 20S proteasomes have been validated as druggable targets for antimicrobials. However, low yields and purity of the native proteasome have hindered studies of the *Tv* 20S proteasome (*Tv*20S). We address this challenge by creating a recombinant protozoan proteasome by expressing all seven α and seven β subunits of *Tv*20S alongside the Ump-1 chaperone in insect cells. The recombinant *Tv*20S displays biochemical equivalence to its native counterpart, confirmed by various assays. Notably, the marizomib (MZB) inhibits all catalytic subunits of *Tv*20S, while the peptide inhibitor carmaphycin-17 (CP-17) specifically targets β2 and β5. Cryo-electron microscopy (cryo-EM) unveils the structures of *Tv*20S bound to MZB and CP-17 at 2.8 Å. These findings explain MZB's low specificity for *Tv*20S compared to the human proteasome and demonstrate CP-17's higher specificity. Overall, these data provide a structure-based strategy for the development of specific *Tv*20S inhibitors to treat trichomoniasis.

*Trichomonas vaginalis (Tv)*, a pear-shaped protozoan organism, is the etiological agent of trichomoniasis, the most widespread non-viral sexually transmitted disease (STD) worldwide[1–3]. This parasite possesses a single flagellum, which enables its motility, as well as several hair-like structures called pili, facilitating its attachment to host cells. *Tv* has a complex cytoskeleton that gives it the ability to alter its shape and traverse host tissues. In women, *Tv* infection can cause vaginitis, while in men it can cause urethritis and prostatitis. Notably, this infection heightens the risk of transmission of HIV and other STDs in both sexes. Current treatment relies on 5-nitroimidazoles[4,5], however,

the emergence of resistant strains poses a significant public health threat due to the lack of alternative treatment options[6,7]. Consequently, effective anti-parasitic compounds are urgently needed. Recently, the critical role of the proteasome in the survival of *Tv* was demonstrated, validating it as a potential drug target for treating trichomoniasis[8–11].

The proteasome is a large protein complex that plays a pivotal role in the degradation and homeostasis of cellular proteins[12–15]. It consists of two main components: the 20S core particle and the 19S regulatory particle. The 20S core particle forms a cylindrical structure

[1]Institute of Organic Chemistry and Biochemistry AS CR, v.v.i., Prague, Czech Republic. [2]Skaggs School of Pharmacy and Pharmaceutical Sciences, University of California San Diego, La Jolla, CA, USA. [3]Department Pharmaceutical Sciences, College of Pharmacy, The University of Jordan, Amman, Jordan. [4]Scripps Institution of Oceanography, University of California San Diego, La Jolla, CA, USA. [5]Department of Medicine, School of Medicine, University of California San Diego, La Jolla, CA, USA. [6]Center for Discovery and Innovation in Parasitic Diseases, Skaggs School of Pharmacy and pharmaceutical Sciences, University of California San Diego, La Jolla, CA, USA. [7]These authors contributed equally: Jan Silhan, Pavla Fajtova. ✉e-mail: fajtova@uochb.cas.cz; ajodonoghue@health.ucsd.edu; boura@uochb.cas.cz

consisting of four stacked rings, comprised of two inner rings of seven different β subunits and two outer rings of seven different α subunits. Importantly, three of the β subunits, β1, β2 and β5, have proteolytic activity and are therefore responsible for protein degradation[16,17].

Structural studies using yeast and mammalian proteasomes have greatly improved the understanding of the mechanisms of protein degradation by these enzymes, as well as their interactions with inhibitors and other proteins. To perform structural studies, highly pure and homogeneous samples are required. We and others have previously isolated proteasomes from parasites such as *Plasmodium falciparum* that are sufficiently pure for structural studies[18,19] but efforts to isolate highly pure proteasomes from *Tv* were unsuccessful[10,20]. One promising strategy to circumvent these technical limitations is to produce recombinant *Tv* proteasome. Archaeal 20S proteasomes, composed of homo-heptameric rings can be produced by co-expressing the two subunits in *E. coli*[21]. However, the tightly regulated biogenesis pathway of eukaryotic 20S proteasome makes expression in bacteria unfeasible[22]. The assembly of the human 20S proteasome involves the stepwise incorporation of 14 distinct protein subunits, α1–α7 and β1–β7, assisted by five dedicated chaperones. These chaperones consist of Ump-1 and the heterodimers, PAC1-PAC2 and PAC3-PAC4[23,24]. The seven α subunits form the α-ring which then serves as a scaffold for the ordered incorporation of β subunits. After dimerization of two pre-assembled half-proteasomes, the final maturation step involves self-cleavage of the β subunit pro-peptides. These N-terminal extensions, present in immature β subunits, are believed to shield their proteolytic activity until formation of the entire 20S proteasome and may also contribute as scaffolds for proteasome assembly[16,25].

Following maturation, the β1, β2, and β5 subunits contain a threonine residue as the N-terminal amino acid. This threonine residue can initiate a catalytic reaction through nucleophilic attack of the carbonyl carbon within the peptide backbone of the substrate. This reaction leads to cleavage of the protein or peptide substrates. The β1, β2, and β5 subunits have distinct substrate specificity preferences and are generally considered to have caspase-like, trypsin-like, and chymotrypsin-like activities, respectively[26]. These three subunits enable the proteasome to cleave a wide variety of substrates at distinct sites allowing the cell to efficiently degrade proteins and maintain cellular protein homeostasis[27,28].

Proteasome inhibitors bind to the proteasome, blocking its proteolytic activity and impeding protein degradation. This results in the accumulation of proteins within cells, ultimately triggering cell death. Proteasome inhibitors have been shown to selectively kill cancer cells and three (bortezomib, carfilzomib, and ixazomib) have been approved for treatment of multiple myeloma and mantle cell lymphoma[29]. Additionally, proteasome inhibitors have recently been investigated as potential treatments for parasitic diseases, such as malaria[19,30,31], Chagas disease and leishmaniasis[32–34], with one molecule, LXE408, developed by Novartis Pharmaceuticals entering Phase II clinical trials for leishmaniasis in December 2022. These clinical studies in different parasites support our rationale for developing proteasome inhibitors to treat trichomoniasis.

Carmaphycin B is a marine cyanobacterial metabolite that was isolated from extracts of *Symploca* sp. collected from Curaçao[35]. This compound is a potent inhibitor of the 20S proteasome from mammals, yeast, trematodes and protozoa[30,35–37] and has potent cytotoxic activity in these cells. We have synthesized more than 100 analogs that exhibit a range of biological activities. Carmaphycin-17 (CP-17) is one analog that has been shown to have reduced cytotoxicity for human cells while exhibiting significant activity against *Tv*[10,35]. Additionally, it has shown to be potent against metronidazole-resistant *Tv* strains[10]. In topical treatment studies of mice vaginally infected with the related trichomonad, *Trichomonas fetus*, CP-17 reduced parasite burden without noticeable adverse effects. These studies not only validated *Tv*

proteasome as a therapeutic target for trichomonacidal agents but also highlighted CP-17 as a starting point for further medicinal chemistry studies[10].

Marizomib (MZB) is a natural product isolated from the marine bacterium *Salinispora tropica*[38]. This non-peptidic proteasome inhibitor is currently in Phase III clinical trials for the treatment of various types of glioblastoma and stands out as the only proteasome inhibitor capable of readily crossing the blood-brain barrier[39–41]. While MZB primarily targets the β5 subunit of the 20S proteasome, it has been reported to also inhibit the β1 and β2 subunits at higher concentrations[42]. This ability to target multiple subunits of the proteasome may contribute to its potent anti-cancer activity. Compared to clinically approved proteasome inhibitors such as bortezomib and carfilzomib, MZB offers advantages including enhanced stability, bioavailability, and the potential for a broader range of anti-cancer activity. While MZB is too toxic for use as an anti-microbial drug, analogs of this molecule could be developed that are selective for *Tv* over mammalian cells.

In this study, we successfully employed the baculovirus expression system to generate recombinant *Tv*20S proteasome. Biochemical comparison with the native enzyme confirmed that all three catalytic subunits of the recombinant enzyme were functional, as they hydrolyzed three different subunit-specific fluorogenic substrates. A broad-spectrum activity-based probe covalently labeled the catalytic threonine from each subunit. Furthermore, we determined the structure of *Tv*20S in complex with MZB and CP-17. MZB was found to bind to six sites within *Tv*20S that corresponded to three subunits in each β-ring, albeit the electron density was insufficient to fully resolve its binding to the β5 subunit. By comparison, CP-17 bound to four of six catalytic subunits, corresponding to β2 and β5 from each β-ring. These findings not only offer valuable insights into the structure and function of *Tv*20S proteasome but also shed light on the underlying molecular mechanisms of MZB and CP-17 inhibition.

## Results

### Expression of recombinant *Tv20S*

The genome of *T. vaginalis* G3 is available from trichdb.org and all 14 subunits of the *Tv*20S proteasome were identified by alignment with the human constitutive proteasome[10] (Supplementary Fig. 1). From these analyses, we identified the putative β7 subunit as A2F3X4 (Uniprot ID) which in the human and yeast proteasome has been shown to be the last subunit incorporated into the α-ring/β-ring half-proteasome[43,44]. A sequence encoding a C-terminal twin-strep tag was added to the β7 subunit (Supplementary Fig. 2) as this was expected to have minimal effect on the assembly of the proteasome complex. In addition, the *Tv* genome was searched for chaperone proteins with homology to PAC1–PAC2, PAC3–PAC4 or Ump-1 that are known to play a role in proteasome assembly in human cells[22,45]. A homolog of human Ump-1 in the *Tv* genome (Uniprot A2FJW0) with 21% sequence identity was identified (Supplementary Fig. 1) however, no homologs of PAC1–PAC2, PAC3–PAC4 were found.

Three baculoviruses were prepared, one bearing seven α subunit genes, one with seven β subunit genes, and a third with the Ump-1 chaperone gene (Fig. 1a). Insect cells were simultaneously infected with the three baculoviruses (Fig. 1b). The resulting recombinant protein complex was enriched from cell lysates on a streptavidin column and further purified by size exclusion chromatography. Only fractions containing catalytically active protein were selected as determined by assays with the Suc-LLVY-amc substrate (Fig. 1c). This substrate is cleaved by the β5 subunits of *Tv*20S that are found on each of the β rings (Fig. 1d). A protein yield of ~1 mg per liter of cell culture was achieved.

#### Protein quality control

To assess the functional similarity of r*Tv*20S and native *Tv*20S (n*Tv*20S), both enzymes were incubated with the fluorescent inhibitor

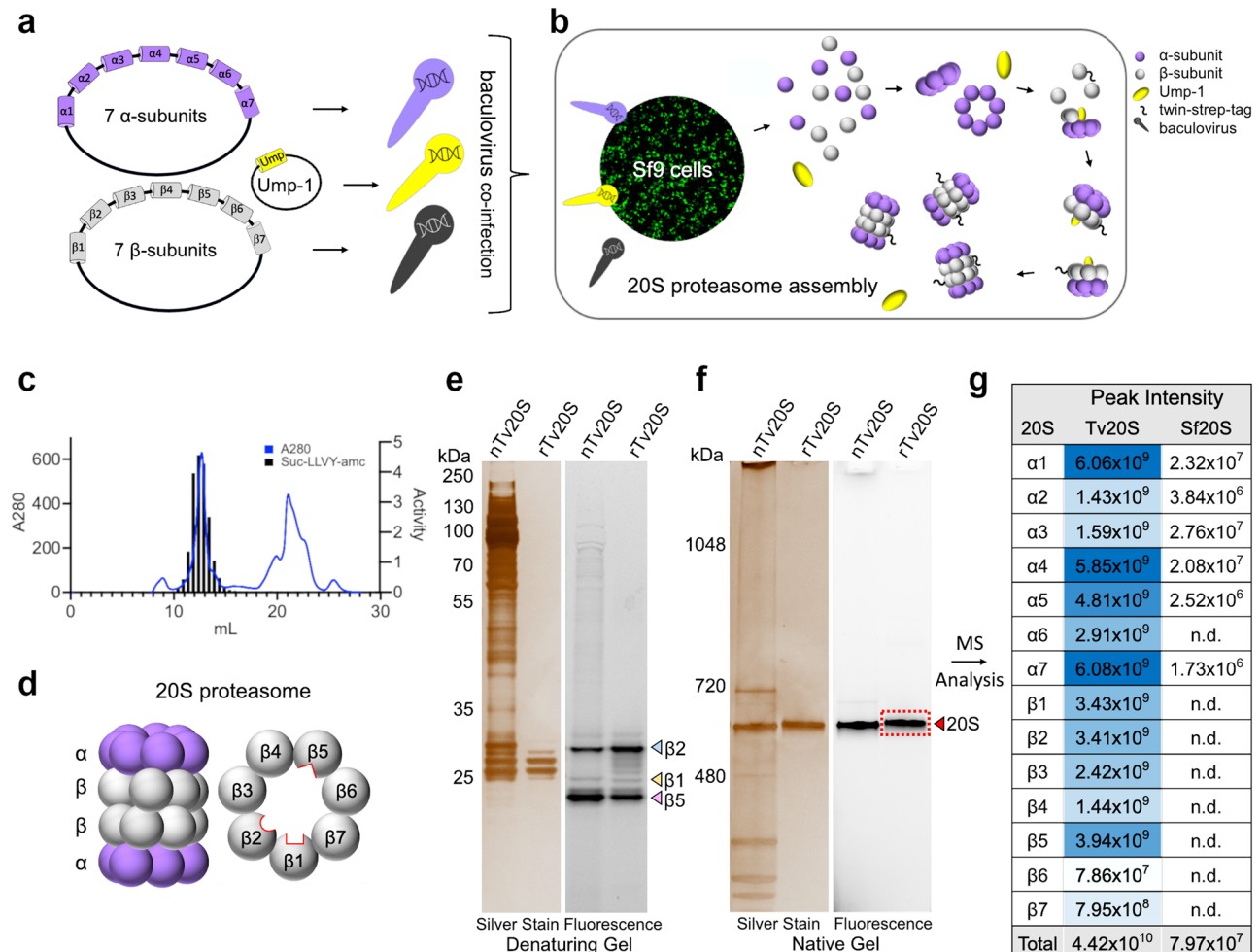

**Fig. 1 | Cloning, Recombinant Expression, and Purification of *Tv*20S Proteasome. a** Cloning of three baculovirus vectors containing either 7 α-subunits, 7 β-subunits or Ump-1 from *T. vaginalis*. **b** Co-infection of Sf9 insect cells with the three baculoviruses for proteasome expression results in assembly of half proteasomes initially and then 20S proteasomes. The twin-strep tag is indicated by ~. **c** Final purification step of recombinant *Tv*20S using Superose 6 chromatography. Absorbance at 280 nm (blue line) and proteasome activity assessed using a fluorogenic substrate, Suc-LLVY-amc (grey bars). **d** Side and planar views of a 20S proteasome complex showing two β rings sandwiched between two α rings. Within

the β ring there are three catalytic subunits (β1, β2, β5) located within the central core. **e** Denaturing gel comparing Me4BodipyFL-Ahx3Leu3VS-labeled n*Tv*20S and r*Tv*20S. The gel was imaged by silver staining and fluorescent scan. **f** Native-PAGE gel of the same proteins from panel **e**. The protein band in the red box was excised for proteomic studies. Gels were repeated independently three times with similar results. **g** Proteomics analysis of the excised band and searched against *T. vaginalis* and *S. frugiperda* proteomes. The blue shading correlates with peak intensity. Source data are provided in the Source Data file.

probe Me4BodipyFL-Ahx3Leu3VS. This probe irreversibly binds to the active sites of β1, β2, and β5 subunits. We then analyzed the labeled enzymes on denaturing and native gels. Consistent with our previous findings on n*Tv*20S[20], the probe strongly labeled the β2 and β5 subunits of both r*Tv*20S and n*Tv*20S, with weaker labeling of β1 subunits (Fig. 1e). On a native gel both enzymes exhibited a similar molecular weight of approximately 690 kDa (Fig. 1f) although r*Tv*20S appeared slightly larger. This could be attributed to the presence of a 28-amino acid Strep-tag on each β7 subunit. This tag alters the overall isoelectric point (pI) of r*Tv*20S and adds approximately 6 kDa in size.

Native *Tv*20S was isolated by sequential steps of ammonium sulfate precipitation, size exclusion chromatography (SEC), and anion exchange chromatography purification while r*Tv*20S was affinity purified of a streptavidin column and then further refined by SEC. To assess the purity of both enzymes, we compared them using silver staining on native and denaturing gels. As shown in Fig. 1e, f, these analyses revealed the high purity of the r*Tv*20S compared to n*Tv*20S.

To understand the role of *Tv* Ump-1, we expressed r*Tv*20S with and without this chaperone protein. In the absence of Ump-1, the

eluted fraction from the streptavidin column contained a higher ratio of half-proteasomes to full-proteasomes, indicating incomplete assembly. Interestingly, incubating these proteins for 72 h at room temperature partially rescued this defect (Supplementary Fig. 3). Additionally, β5 subunit activity, measured using both the fluorogenic probe and substrate, was significantly weaker when Ump-1 was absent (Supplementary Fig. 3). The connection between Ump-1 and β5 activity is supported by previous studies of the yeast proteasome where Ump-1 was shown to directly interact with the propeptide sequence of β5 and properly orientate it between the β6 and β7 subunits[46]. These findings collectively demonstrate that recombinant *Tv* Ump-1 functions as a critical chaperone, essential for the complete maturation and full activity of r*Tv*20S.

Since r*Tv*20S was expressed in *Spodoptera frugiperda* insect cells, we investigated the possibility of contamination with insect proteasome subunits. While the overall size difference allowed clear separation of r*Tv*20S and the host cell proteasome (*Sf*20S) on a native gel (Supplementary Fig. 3), the potential incorporation of individual insect subunits into r*Tv*20S remained a concern. To address this, the r*Tv*20S

band from the native gel was excised and analyzed by proteomics. All 14 *Tv* subunits were identified and quantified when the mass spectrometry data was searched against the *Tv* proteome (Fig. 1g). Additionally, the data was searched against the insect proteome. While six insect subunits were detected, their abundance was significantly lower (>100-fold) than the equivalent *Tv* subunits (details are provided in the Source Data file). These quality control measures demonstrate that r*Tv*20S is a highly pure and enzymatically active enzyme complex, suitable for further biochemical and structural studies.

## Biochemical characterization of r*Tv20S*

We next compared the catalytic activity of the r*Tv*20S and n*Tv*20S using a combination of inhibitors and fluorogenic substrates. As revealed in Fig. 1e, the fluorescent probe, Me4BodipyFL-Ahx3Leu3VS, binds to all three catalytic subunits. Preincubation of the proteasome with irreversible-binding inhibitors prevents labeling with the probe. When r*Tv*20S and n*Tv*20S were incubated with 50 μM of CP-17 followed by the probe, the β2 and β5 subunits were no longer labeled (Fig. 2a). When the same proteins were incubated with 50 μM MZB none of the catalytic subunits were labeled with the probe. These data reveal that CP-17 binds to β2 and β5, while MZB binds to all three catalytic subunits (Fig. 2b). In addition, these studies show that r*Tv*20S and n*Tv*20S have the same inhibition profile when using Me4BodipyFL-Ahx3Leu3VS as a reporter of subunit targeting. As a control, human 20S (*Hs*20S) was preincubated with CP-17 and MZB prior to labeling with the probe. CP-17 completely inhibited β5, partially inhibited β2 and did not inhibit β1, while MZB targeted all three subunits.

The functionality of r*Tv*20S was further compared to n*Tv*20S through enzyme kinetic assays using subunit-specific fluorescent substrates[20]. Using a concentration range of 2 to 250 μM for the β2 and β5 substrates and 10 to 1250 μM for the β1 substrate, the specific activity was indistinguishable between the two enzymes, and the catalytic rate constant ($k_{cat}$) and Michaelis constant ($K_M$) were calculated for all three subunits of r*Tv*20A and n*Tv*20S (Fig. 2c) to reveal that these enzymes were equivalent.

Finally, we used fluorogenic assays to show that the relative activity of n*Tv*20S and r*Tv*20S in the presence of inhibitors is similar. Using the β5 and β2 substrates, the activity of both enzymes was reduced to comparable levels following pre-incubation with 50 μM CP-17 or MZB (Fig. 2d). For the β1 substrate, MZB reduced activity for both enzymes to comparable levels. However, CP-17 activated the β1 activity of n*Tv*20S by 200% but activated r*Tv*20S by only 120%. Although activation of β1 in the presence of β5 and β2 inhibitors was observed previously for n*Tv*20S[20] it is unclear why this occurs. Our follow-up studies revealed that β1 activation of r*Tv*20S in the presence of CP-17 is independent of the substrate concentration used in the assay with activation ranging from 120% to 160% (Supplementary Fig. 4). In summary, these probe-labeling and enzyme activity assays revealed that r*Tv*20S is enzymatically identical to n*Tv*20S and therefore, validates the use of the recombinant enzyme for biochemical and structural studies.

## Overall structural characterization of the *Tv20S* proteasome

To obtain atomic models of the *Tv*20S proteasome structure with bound inhibitors, we mixed r*Tv*20S with a 50-fold excess of CP-17 or MZB and subjected it to cryo-EM. Initial data analysis revealed that the recombinant sample with MZB contained a significant amount of unassembled proteasome. During the 2D classification of the collected dataset, only 2% of the particles corresponded to the assembled 20S proteasome, while the remaining ring particles mostly consisted of half-proteasome complexes made up of a single α and β ring. The low abundance of full proteasomes was not due to inhibitor-dependent disassembly of proteasome (Supplementary Fig. 3e) and therefore may have occurred during the grid set-up.

We were able to obtain a 2.86 Å cryo-EM map of the *Tv*20S proteasome, which was used to build the initial atomic model of the r*Tv*20S with MZB; PDB ID: 8OIX. For r*Tv*20S bound to CP-17, we performed a second round of gel filtration to enrich the larger 20S proteasomes over the half-proteasomes. Each fraction was then analyzed for the presence of fully assembled *Tv*20S using negative-stain electron microscopy. This procedure improved yields so that approximately 80% of all particles could be assigned to the fully assembled *Tv*20S proteasomes. The data collected from this sample allowed the cryo-EM reconstruction and determination of the atomic structure of *Tv*20S with CP-17; PDB ID: 8P0T.

The initial *Tv*20S atomic model was built by leveraging AlphaFold models, which were superimposed onto the existing 20S proteasome structure of *Leishmania tarentolae* 20S proteasome (PDB ID: 7ZYJ)[47] and fitted to the cryo-EM maps. Subsequently, the model was refined and rebuilt based on these maps. Unsurprisingly, *Tv*20S exhibits a shape and fold consistent with the characteristic overall structure of 20S proteasomes. *Tv*20S displays C2 symmetry, with the two sets of 14 subunits arranged in the conventional ring configuration of α1–α7, β1–β7 / β1–β7, α1–α7. Despite the homology among the individual chains between *Tv* and *L. tarentolae* 20S proteasome variations in their arrangement were discovered. The overall fold of the *Tv*20S proteasome somewhat differs from other proteasomes with known structures (e.g., RMSD = 2.729 Å for Cα atoms in structural alignment with the *L. tarentolae* 20S proteasome). It is worth noting that, in these structures, the *Tv*20S sequence (3167 residues) was mapped onto the cryo-EM maps, achieving 96.8% coverage for *Tv*20S/MZB and 96.6% for *Tv*20S/CP-17 structures. This includes well-defined regions, such as the active site pockets, which allowed for the modeling of inhibitor molecules in both structures.

## Structure of active sites of the catalytic subunits β1, β2, and β5 bound to inhibitors

All three active sites in *Tv*20S are defined by a conserved catalytic triad Thr1, Asp17, and Lys33[26]. In addition, other well-conserved residues, such as Asp168, Ser131, and Ser170 in β1, are predicted to be required for efficient catalysis and active site structural integrity[48]. In the *Tv*20S-MZB dataset, the electron map within the β1 active site exhibited clear density extending beyond Thr1, providing sufficient coverage for the inhibitor MZB. As a result, the atomic model of MZB was successfully built and fit accurately into the cryo-EM map (Supplementary Fig. 5). Conversely, no density was observed within the β1 active site for CP-17 in the other dataset. This is consistent with the biochemical data (Fig. 2d), which demonstrated that MZB inhibits β1 activity while CP-17 does not.

The second active subunit, β2, can accommodate both MZB and CP-17 inhibitors in its active site, as clearly shown by the cryo-EM maps of those regions (Supplementary Figs. 5 and 6). MZB adopts a conformation like that observed in the β1 site. In both cases, the cyclohexenyl moiety of MZB and the phenyl residue of CP-17 occupy a deep pocket that is shared by several residues, including Lys33, and a loop composed of Ala46, Ala49, Glu31, and Asn52 (Fig. 3f). This pocket is commonly referred to as the S1 pocket of the enzyme because it serves as the binding site for the P1 amino acid of the substrate, which is on the N-terminal side of the scissile bond. For CP-17, the α,β-epoxyketone group is covalently bound to Thr-1 forming a morpholino derivative. Beyond the S1 pocket, the P2 tryptophan of CP-17 interacts with the β2 active site through its indole ring that resides on the hydrophilic ridge of the shallow S2 pocket, whereas the indole ring of the P3 tryptophan is buried and fits well within the deep S3 pocket. The S3 site is characterized by the presence of aliphatic chains, including Val28 and six alanine residues (Ala20, Ala22, Ala27, Ala124, Ala130, Ala132). Additionally, Glu31 and Asp126 line the S3 pocket, creating an acidic environment. The N-terminal capping group of CP-17 is a hexyl chain, which is orientated in a shallow S4 pocket.

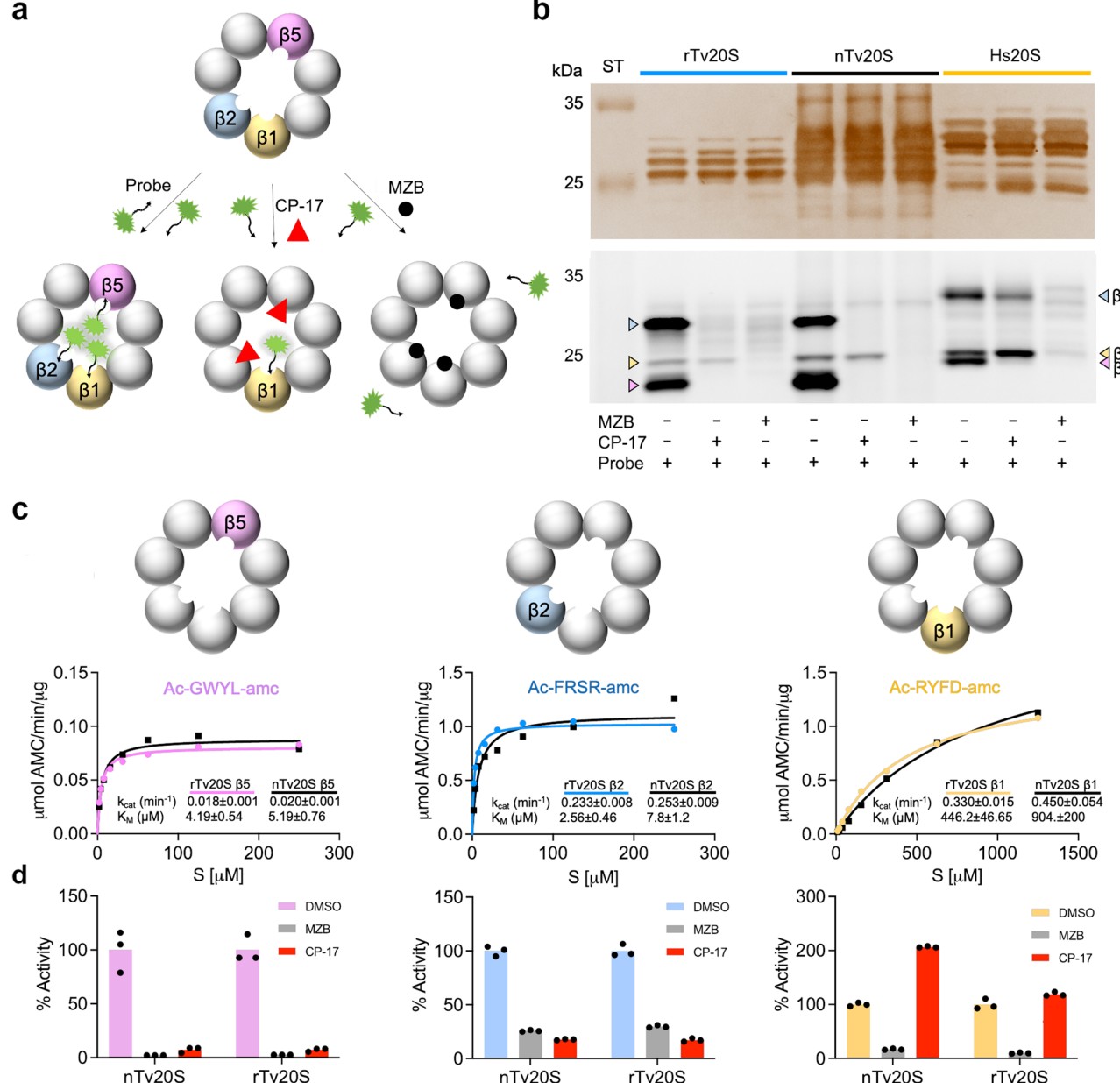

**Fig. 2 | Biochemical and enzymatic characterization of recombinant *Tv*20S.** **a** Schematic representation of active sites during inhibition by MZB or CP-17, followed by visualization through fluorescent Me4BodipyFL-Ahx3Leu3VS probing. The inhibitor is depicted binding to specific active sites, thereby blocking enzymatic activity. Subsequent visualization techniques provide insights into the binding interaction and potential conformational changes induced by the inhibitor. **b** r*Tv*20S, n*Tv*20S and *Hs*20S were first incubated for 1 h with 50 μM CP-17 or 50 μM inhibitor or the solvent control, DMSO, and then incubated with 2 μM Me4BodipyFL-Ahx3Leu3VS for 16 h, followed by fractionation on a denaturing gel and fluorescence imaging at 470 nm excitation and 530 nm emission. The experiment was repeated independently three times with similar results. 'ST' stands for the molecular weight ladder. **c** Comparison of dose response curves for n*Tv*20S and r*Tv*20S with $K_M$ and $k_{cat}$ values indicated. Assays were performed in triplicate, with each dot representing the mean of 3 technical replicates. **d** Comparison and validation of inhibitor specificity using the indicated subunit-specific *Tv*20S proteasome substrates. Assays were performed in technical replicates ($n = 3$).

In the catalytic site of β5, electron density corresponding to MZB was observed in the cryo-EM map. However, unlike the observations made for the β1 and β2 sites, the MZB in β5 was not sufficiently defined to allow for a satisfactory fit. Even though we found that MZB inactivates the β5 catalytic activity, it could not be included in the final model. In contrast, the density of CP-17 in the β5 site was clearly defined. The ligand was positioned in a highly similar conformation to that observed in the β2 site (Fig. 3). The α,β-epoxyketone group was covalently bound to Thr-1 and the P1 phenylalanine is located in the S1 pocket, which is lined with Lys33, Ala46, and Ala69. In contrast to the β1 and β2 sites, the S1 of β5 harbors Val31 and Met35. Moving forward, the P2 indole ring is

positioned on the ridge of S2, formed by the loop between Ala46 and Ala49, and is enclosed by Ser96. The P3 indole is anchored in the relatively deep S3 pocket, defined by π–π interaction with the residue Phe27. Finally, the hexyl group of CP-17 sits in a well-defined shallow pocket formed by the hydrophobic parts of four residues belonging to the neighboring β6 chain (Tyr116, Asp136, Pro137, and Val138).

## Comparison of the active proteasome sites of human and *T. vaginalis* for improved design of selective inhibitors
At first glance, the overall fold of the human constitutive proteasome and *Tv*20S appears very similar. However, upon structural alignment,

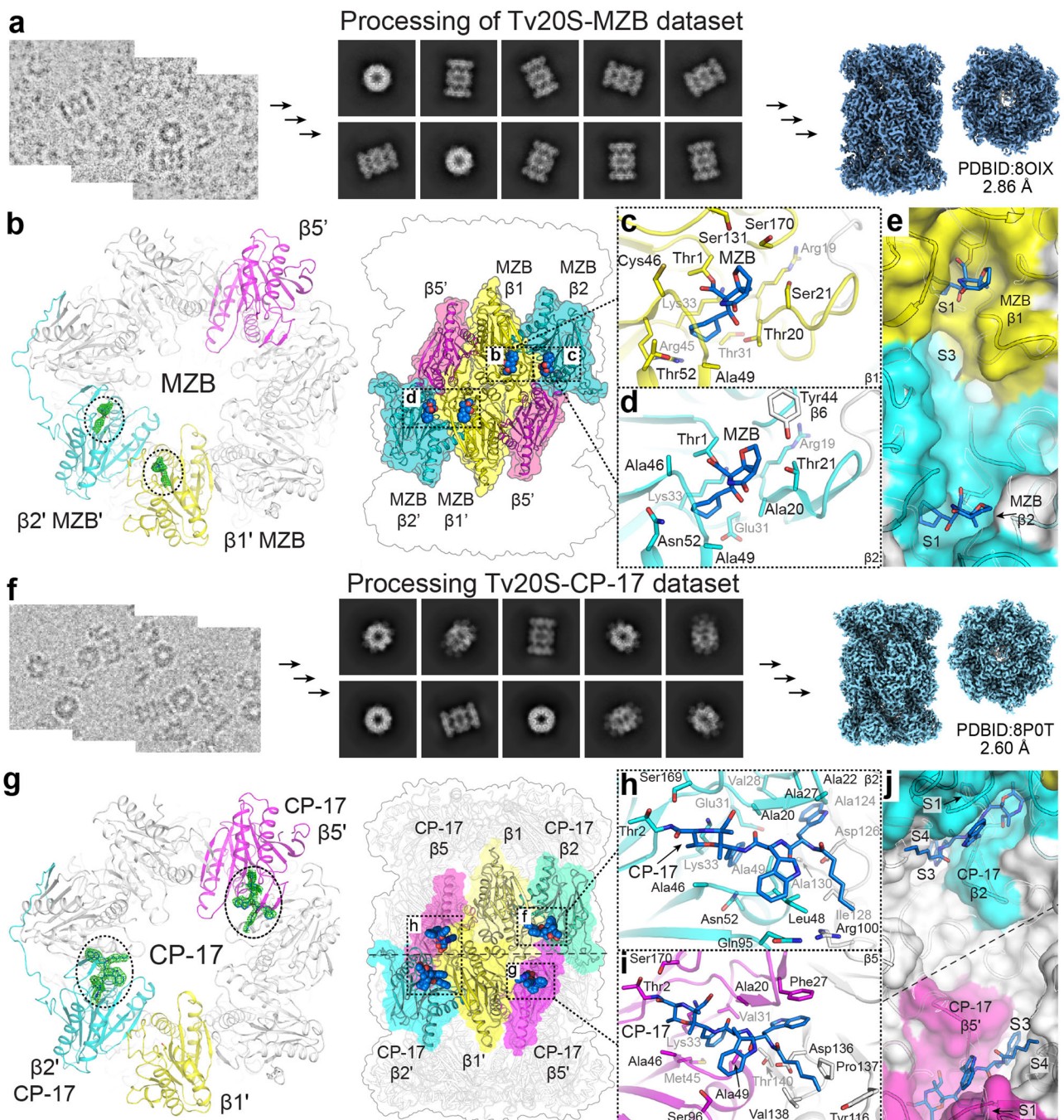

**Fig. 3 | Cryo-EM structures of *Tv*20S bound to inhibitors.** Processing of the cryo-EM datasets for **a** *Tv*20S bound to MZB and **f** CP17. The electron maps highlight the inhibitors in the active sites (**b**), **g**, shown as green mesh), while the catalytic subunits are depicted in different colors (β1 yellow, β2 cyan, and β5 magenta). MZB was observed only in the catalytic sites of β1 and β2 (**b**–**e**), while CP-17 was observed only in the catalytic sites of β2 and β5 (**g**–**j**). Both inhibitors were covalently bound to Thr1 as detailed for MZB in panels **b**–**e**) and for CP-17 in panels **g**–**j**. Amino acid residues within a reach of non-covalent interactions (up to 4 Å) are shown as sticks. More detailed interactions between inhibitors and active site pockets are shown in Supplementary Figs. 12–14.

the overall root mean square deviation (RMSD) is calculated to be 2.638 Å (Fig. 4a), which is higher than anticipated given the conservation of individual subunits. This disparity can be attributed to the slightly different packing of *Tv*20S, leading to less-than-optimal overall alignment. When aligning the individual subunits, the RMSD values decrease accordingly as the aligned sequences exhibit a range of 25–51% amino acid identity, with conserved regions in the active sites (Supplementary Table 1). Despite this relatively high sequence identity, the fine structural variances within the active site pockets present

opportunities for designing selective inhibitors targeting *Tv*20S over human 20S, as illustrated by CP-17, which inhibits the *Tv*20S β5 subunit with a 10-fold higher potency than the equivalent subunit of the human constitutive proteasome[10]. In the human 20S proteasome, several amino acids pose steric hindrances, resulting in clashes when hypothetically positioning the CP-17 inhibitor within the active sites based on the alignment of *Tv*20S and the human proteasome.

In the β2 active site, many of the residues are conserved but in the S3 pocket where the second indole ring is positioned, the human 20S

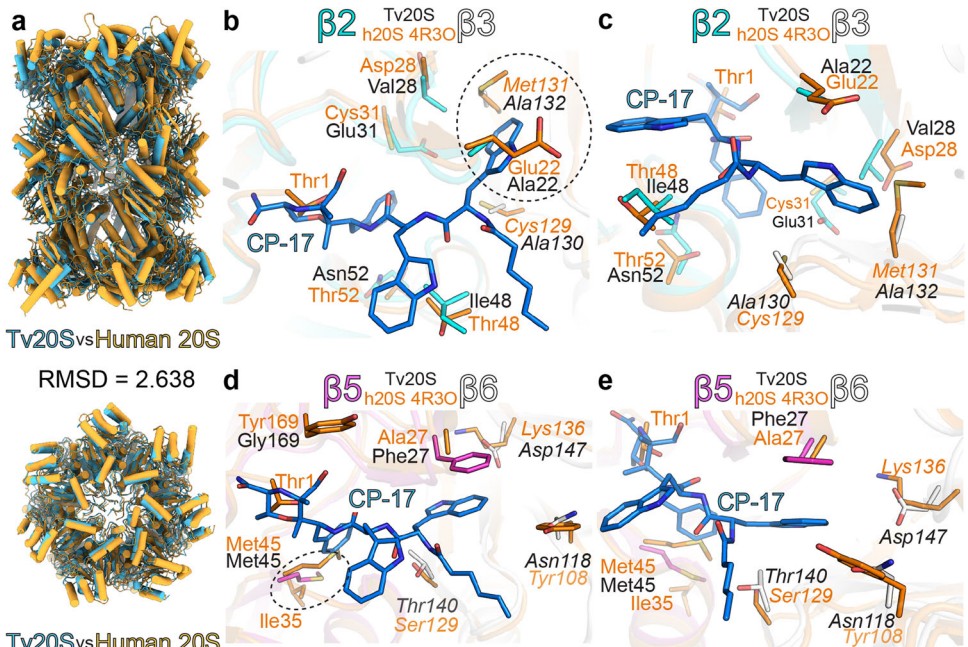

**Fig. 4 | Structural comparison of *Tv*20S and human 20S.** The panel **a** comparison of human (orange) and *T. vaginalis* (light blue) 20S proteasomes. The further panels (**b**–**d**) show structural overlay of human 20S and covalent inhibitor CP-17 (blue sticks) in the active site pockets of β2 (panels **b**, **c**) and β5 (panels **d**, **e**) of *Tv*20S. The residues responsible for major structural differences in these active sites are shown as sticks (human-orange, *Tv*20S β2 cyan, and β5 magenta, and other subunits shown in white). The predicted clashes between CP-17 and amino acids from human 20S are encircled with dashed lines. In panels **b**, **c**, Met131 of human β2 is in clash with one of the indole rings of CP-17, whilst in *Tv*20S this moiety is accommodated in a deep hydrophobic pocket formed by Ala132 and Ala130. Similarly, in panels **d**, **e**, the pocket formed by Met45 of *Tv*20S β5 allows sufficient space for the phenyl ring of CP-17. In this location, human β5 Met45 is pushed by Ile35 towards the active site, forming a potential barrier for CP-17. The further detailed views highlighting the differences between human 20S and *Tv*20S-CP-17 in particular are shown in Supplementary Fig. 15.

contains bulkier residues, such as Met131 and Ser122, in contrast to the significantly smaller alanine residues found at these positions in the *Tv*20S β2 site (Fig. 4b, c). The S3 pocket of the β5 subunit of *Tv*20S contains Phe27, which forms a favorable π-π interaction with the indole ring and represents the sole significantly different residue in this site. Conversely, the S1 pocket of the human active site contains Met45 and Ile35, which may obstruct the binding of the phenyl ring (Fig. 4d, e). It is plausible that Met45 from *Tv*20S exhibits greater mobility, moving away from the approaching ligand and not being hindered by Ile35, while the π-π interaction with Phe27 enhances the binding of the ligand in the case of *Tv*20S proteasome. This differential binding mechanism allows CP-17 to bind to the β5 site with greater selectivity compared to the human enzyme.

Furthermore, upon exploring the surfaces and binding interfaces of all the active sites in *Tv*20S and human 20S proteasomes, it becomes evident that the β2 and β5 active sites, along with their respective S1 and S3 pockets, exhibit significant differences in shape. These differences offer additional opportunities for further inhibitor design (Fig. 5). Although the distinct amino acid composition and slight differences in the arrangement of the β2 compared to β5 active sites pose challenges for the design of selective β2 inhibitors.

## Discussion

While structural and enzymatic studies are crucial for identifying exploitable differences between parasite and human proteasomes, research has been hampered by difficulties in acquiring sufficient enzymes. Isolating proteasomes from intracellular parasites like *Plasmodium* and *Babesia* is challenging due to their complex cultivation needs. Even for parasites with easier culturing, such as *Trypanosoma*, *Leishmania*, and *Trichomonas*, obtaining highly purified enzyme remains difficult. For instance, our recent work revealed co-purification of actinin and a legumain protease with native *Tv*20S[10,20].

This study presents a powerful solution, utilizing a heterologous expression system to overcome purity and yield limitations. We report a pathogen proteasome expressed in an insect cell line. Three separate baculoviruses containing the *Tv* genes for 7α, 7β, and Ump-1 were used to infect Sf9 cells, resulting in the production of fully functional recombinant *Tv*20S. This was also confirmed by the observation of fully assembled *Tv*20S with the C-terminus of the β7 subunit, that is known to stabilize nascent 20S[49], properly inserted into the space between β1 and β2 of the opposing half proteasome (Supplementary Fig. 7). *Tv*20S was an ideal candidate due to our existing in-depth understanding of its catalytic subunits and the ability to directly compare the native and recombinant enzymes[20]. Maintaining consistency between native and recombinant *Tv*20S in inhibitor interactions is critical to avoid misinterpretations in structure-activity relationship studies.

The successful expression of *Tv*20S using the baculovirus system suggests broad applicability to proteasomes from other pathogens. For intracellular parasites like *Plasmodium falciparum* and *Babesia divergens*, parasitemia levels are often low, and therefore obtaining high yields of enzyme is difficult[30,31,50]. Many pathogens don't replicate in culture at all, necessitating harvest from invertebrate animals. For instance, the life cycle of *Schistosoma mansoni* requires both freshwater snails (intermediate host) and hamsters. We previously isolated the 20S proteasome from *S. mansoni* worms, demonstrating its potential as a drug target[36]. Therefore, a method for expressing recombinant 20S proteasomes from various pathogens would significantly aid biochemical and structural studies.

Our purified recombinant *Tv*20S was used to determine two structures in complex with different inhibitors. Despite the overall structural conservation of proteasomes across all life forms (archaea to humans), *Tv*20S exhibits some key variations. Like other eukaryotes, *Tv*20S possesses four stacked heptameric rings. The outer rings

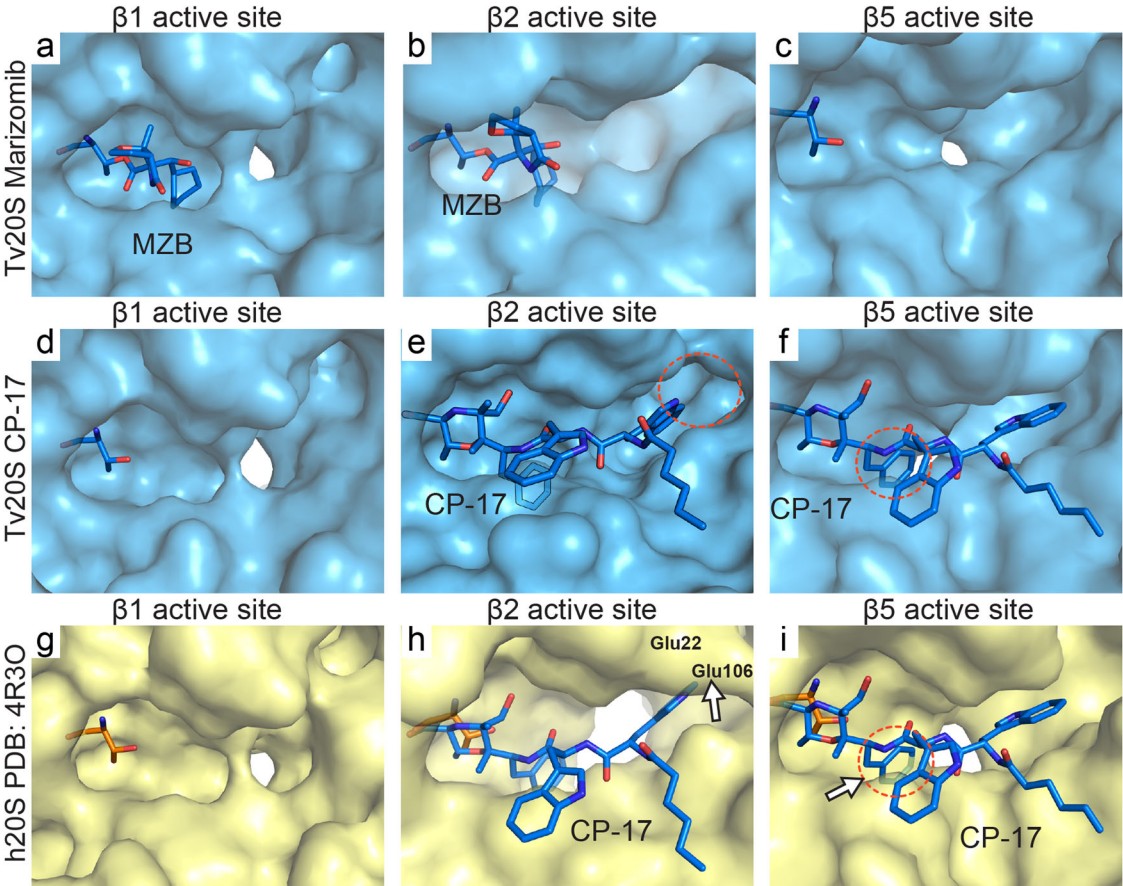

**Fig. 5 | Surface detail of comparison of *Tv*20S and human 20S active sites.** *Tv*20S surface is in light blue, panels **a**–**f**, and its Thr1 residue is shown as blue sticks. *Hs*20S proteasomes is in light yellow, panels **g**–**i**, and its Thr1 is shown as orange sticks. MZB and CP-17, shown as blue sticks, are displayed in the human active site pockets using a structural overlay from Fig. 4. The clashes are encircled with a red dotted line. In panel **h**, Met131 of human β2 forms shallower binding pocket in this region whilst a similar elevation of active site pocket is observed formed by Met45 from human β5. In this location, human Met45 (β5) in this ligand-free structure is pushed by Ile35 towards the active site. Please refer to Supplementary Fig. 15 for more structural detail highlighting these differences including the architecture of human Glu22 (β2), and Glu106 (β3).

consist of α subunits for scaffolding, while the inner rings contain the catalytic β subunits. Unlike archaea where these subunits are identical, *Tv*20S β subunits have diverged while maintaining high sequence similarity with other eukaryotic pathogens (Supplementary Fig. 1). Using sequence homology to human counterparts, we identified each *Tv*20S α and β subunit, along with a homolog of human Ump-1 in the *Tv* genome. Functional assays confirmed the activity of all three catalytic subunits as they were labeled by an activity-based probe and could cleave previously established substrates.

Furthermore, we investigated the effect of the proteasome inhibitor CP-17 on both native and recombinant *Tv*20S. Interestingly, CP-17 inhibited the β2 and β5 subunits in both enzyme forms, while β1 activity was surprisingly enhanced. These findings demonstrate a high degree of consistency between the native and recombinant enzyme in inhibitor interactions, bolstering confidence in the recombinant *Tv*20S as a valuable tool for future drug development endeavors.

Data from our structural studies reveal how MZB and CP-17 inhibit *Tv*20S and provide a structural basis for the development of more specific inhibitors. For instance, the arrangement of the S2 substrate binding pocket in the β1 subunit prevents interactions with CP-17. This binding pocket contains Pro27, Gln112 and Gln127. The two Gln residues are responsible for narrowing the binding grove which prevents binding of the P2 indole ring of CP-17. The active site pocket that accommodates the cyclohexenyl group of MZB appears to be too shallow to accommodate the phenyl group of CP-17. In the overlay of

these β1 and β2 subunits, residue Arg45 of the β1 pocket is too close to phenyl group (2.4 Å) (Supplementary Figs. 8 and 9).

Our studies revealed that MZB inhibits the three catalytic subunits of Tv20S, therefore binding to all six active sites. In contrast, CP-17 selectively inhibits β2 and β5, leaving two active sites available. Interestingly, enzyme activity assays showed that CP-17 inhibition of β2 and β5 subunits led to increased substrate cleavage by the remaining β1 subunit (Fig. 2d and Supplementary Fig. 4). To investigate this unexpected activity increase, we compared the structures of the β1 subunit bound to MZB and unbound in the CP-17 complex (Supplementary Fig. 9). We hypothesized that the unbound β1 subunit might have a conformation that facilitates substrate access. However, both structures (empty and MZB-bound) displayed identical shapes and volumes (Fig. 5, d), suggesting a structural change was not responsible for the increased activity. Therefore, we propose that the higher activity is due to increased substrate concentration at the β1 subunit. With β2 and β5 blocked by CP-17, the substrate has only β1 available for interaction, leading to a higher observed activity compared to the uninhibited enzyme.

CP-17 demonstrates promising potential as a candidate for trichomoniasis by targeting *Tv*20S proteasome active sites with high potency and selectivity. However, the structural knowledge gained here will facilitate the design of now analogs with even greater potency and selectivity. An ideal inhibitor would, additionally, bind the β1 site of *Tv*20S but not the β1 subunit of the human proteasome. Future CP-17 analogs could exploit the presence of Cys46 in *Tv*20S's β1 S1 pocket

(Supplementary Fig. 8a) to form a covalent bond with this residue, potentially enhancing selectivity for Tv20S. This is particularly attractive because the human 20S proteasome lacks Cys46 (Supplementary Fig. 1).

In summary, recombinant proteasome expression via a baculovirus system offers a powerful tool for studying parasite proteasomes. Despite the overall structural similarity, key variations in Tv20S compared to human 20S enable selective targeting. Inhibitor studies with Tv20S revealed a mechanism for increased β1 activity with CP-17, and identified Cys46 in the β1 subunit as a promising target for specific drug development.

## Methods

### Cloning, expression, and protein purification

The cDNA sequences of the 14 proteasome subunits and the assembly chaperone (maturation factor) Ump-1 were identified for the *T. vaginalis* G3 strain on TrichDB by aligning them with human proteins (Supplementary Fig. 1). Required genes were *E. coli* codon-optimized, synthesized (Azenta) and cloned into three separate pACEBac1 vectors by restriction cloning (Supplementary Fig. 2). The plasmids were transposed to DH10EmBacY cells (Geneva Biotech) and the baculoviruses and protein expression was performed in Sf9 insect cells (Thermo Fisher, catalog number #11496015). according to the manufacturer's instructions.

The expression of the 20S proteasome was performed in the presence or absence of the Ump-1 chaperone to understand its importance. Sf9 cells were collected after 72 h postinfection and centrifuged for 15 min at 2,000 x g and the pellet was diluted in Buffer A (50 mM Tris pH 7.5, 150 mM NaCl, 1 mM dithiothreitol, 1 mM EDTA) at a volume corresponding to 4-times the volume of the pellet. Cells were lysed by sonication and the lysate was cleared by centrifugation at 30,000 g for 20 min. The cleared lysate was loaded onto tandem Streptactin XP columns (QIAGEN), equilibrated in Buffer A. Upon extensive washing with Buffer A, protein was eluted in the same buffer supplemented with 50 mM biotin. Proteasome-containing fractions were pooled, concentrated using ultrafiltration (Amicon) and loaded onto a Superose 6 Increase 10/300 gel filtration column (GE Healthcare) equilibrated with 50 mM HEPES pH 7.5, 150 mM NaCl, 1 mM EDTA. Fractions of 0.5 ml were obtained and assayed for protease activity. Enzymatically active fractions were pooled and concentrated to 1 mg/ml. The native Tv20S proteasome was purified from frozen pellets of *Tv* parasites as described previously[20].

### Proteasome activity and inhibition assays

The β5 proteolytic activity of the 20S proteasome was measured in insect cell extracts and in fractions isolated from chromatography columns using the fluorogenic substrate, Suc-LLVY-amc (Cayman Chemical). For enzyme characterization studies, substrates specific for Tv20S β1, β2, and β5 subunits, namely Ac-RYFD-amc, Ac-FRSR-amc, and Ac-GWYL-amc, were used as described previously[20]. These substrates were custom-synthesized by GenScript, NJ. For inhibition studies, 1 µM recombinant Tv20S was incubated with 50 µM MZB, 50 µM CP-17 inhibitor, or 0.5% DMSO (vehicle control) for 1.5 h at room temperature. The inhibition of individual Tv20S subunits was confirmed using the subunit specific fluorogenic substrates. Kinetic assays in 384-well plates were performed using 4.4 nM recombinant Tv20S (rTv20) or enriched native Tv20S (nTv20S) in 50 mM HEPES pH 7.5 with 80 µM of Ac-RYFD-amc, 25 µM of Ac-FRSR-amc, and 25 µM of Ac-GWYL-amc, in a final volume of 8 µL per well. All assays were performed in triplicates in 384-well black plates (Nunc) at 37 °C using a Synergy HTX Multi-Mode Microplate Reader (BioTek, Winooski, VT) with excitation and emission wavelengths of 360 and 460 nm, respectively.

### Gel-based active site probing

Protein lysates, nTv20S, rTv20S, Hs20S, and 20S-inhibitor complex were diluted with 50 mM HEPES pH 7.5 then mixed with 2 µM Me4BodipyFL-Ahx3Leu3VS (R&D Systems #I-190). After probe addition, samples were incubated at room temperature for 16 h. For denaturing gels, samples were mixed with 4X Bolt LDS sample buffer (Thermo Fisher Scientific) containing 250 µM dithiothreitol, heated at 100 °C for 5 min, and loaded onto a NuPAGE 12% Bis-Tris gel (Thermo Fisher Scientific, P/N). PageRuler Plus pre-stained protein ladder (Thermo) was included on each gel. Gels were run with 1X MOPS SDS buffers (Invitrogen) at 130 V. For native gels, samples were mixed with 2X Novex Tris-glycine native sample buffer and loaded onto NuPAGE 3-8% Tris-glycine gels (Invitrogen) with NativeMark unstained protein standard (Thermo Fisher Scientific, P/N 57030). Gels were run at 100 V with Novex Tris-glycine running buffer (Invitrogen). All gels were imaged on Bio-Rad ChemiDoc XRS+ at 470 nm excitation and 530 nm emission for Me4BodipyFL-Ahx3Leu3VS probe visualization and silver stained.

### Native gel digestion and desalting

After the native gel was silver stained, the band with recombinant proteasome (n = 1) was excised, sliced into smaller fragments, and washed three times with 25 mM NH$_4$HCO$_3$, 50% acetonitrile for 10 min each time. Next, the band pieces were dried completely in a Savant Speed Vac Plus AR (Thermo Fisher Scientific). Reduction and alkylation were performed by adding a mixture of 10 mM TCEP and 25 mM iodoacetamide in 25 mM NH$_4$HCO$_3$ to the gel pieces and this reaction proceeded in the dark for 1 h, followed by a 25 mM NH$_4$HCO$_3$ wash and subsequent 25 mM NH$_4$HCO$_3$, 50% acetonitrile wash to dehydrate the gel. The sample was then dried in a Savant Speed Vac Plus AR and 12.5 ng/µL trypsin in 25 mM NH$_4$HCO$_3$ was added. The sample was incubated at 4 °C before being covered in 25 mM NH$_4$HCO$_3$ where the digestion proceeded at 37 °C for 20 h. The supernatant was transferred to a clean tube and the remaining peptides were extracted from the gel by the addition of 50% acetonitrile, 5% formic acid. The extracted digest was dried and resuspended in 0.1% formic acid before C18 ZipTip desalting. A C18 column was washed with methanol and spun for 45 s at 3500 x g. The column was equilibrated with 0.1% formic acid in 50% acetonitrile followed by 0.1% formic acid in water. The sample was loaded onto the column and spun for 2 min at 2000 x g. It was then washed with 0.1% formic acid then eluted from C18 with 50% acetonitrile, 0.1% formic acid by spinning at 3500 x g for 45 s. The sample was dried in a Savant Speed Vac Plus AR and stored at −80 °C for 1 week prior to preparing it for mass spectrometry.

### LC-MS/MS

The sample was redissolved in 0.1% formic acid prior to LC-MS/MS injection. Chromatography was performed on an Easy-nLC 1200 (Thermo Fisher Scientific), with a 75 µm i.d. PicoFrit (New Objective, Woburn, MA) column packed with 1.9 µm AQ-C18 material (Dr. Maisch, Germany) to 50 cm in length. Peptides were separated at 50 °C over 46 min run which consisted of 6% solvent B at 1 min to 30% B in 21 mins, 72% B by 30 min, 90% by 31 min, and finally 60% B until 46 min. Mass spectrometry was performed on a Orbitrap Eclipse with ETD and PTCR (Thermo Fisher Scientific). The scan range was 350–1800 m/z, resolution of 60,000, and a 50 ms maximum injection time. The top 8 scans were selected for MS2. MS/MS spectra were analyzed in PEAKS Studio (v 8.5) software (Bioinformatics Solutions Inc.). MS2 data were searched against the *T. vaginalis* and *S. frugiperda* proteome (UniProt taxon ID: 412133, ID: UP000829999). A precursor tolerance of 20 ppm and 0.01 Da was defined. Trypsin digestion was specified. The data can be found at ftp://massive.ucsd.edu/v07/MSV000094569/ or at proteome Xchange: PXD051584.

## Preparation of cryo-EM grids and data acquisition

Recombinant purified *Tv*20S (1.4 µM) was mixed with 1.25 mM of either MZB or CP-17 to achieve a final concentration of 50 µM inhibitor. The mixture was prepared in a solution of 50 mM HEPES pH 7.5 and incubated at room temperature for 1 h. After the incubation period, the sample was cooled on ice. Simultaneously, the cryo-EM Quantifoil R2/1 300-mesh copper grids (EM Sciences, Prod. No. Q350CR1) were glow discharged at 15 mA for 30 seconds to enhance their hydrophilicity. The proteasomes-inhibitor complex (4 µL) at a concentration of 1 mg/ml was transferred to the grid. The grids were blotted at −5 power for 5 s in FEI Vitrobot Mark IV (Thermo Fisher Scientific) at 4 °C and 100% humidity and immediately frozen in liquid ethane. Immersion freezing and screening of cryo-EM data and all data acquisition were performed at the Umeå Core Facility for Electron Microscopy, Sweden. Screening was performed using a 200 kV Glacios system (Thermo Fisher Scientific) equipped with a Falcon 4i direct electron detector. 7983 images of *Tv*20S with MZB and 10,037 images of *Tv*20S with CP-17 were acquired with a pixel size of 0.7 Å with an exposure of 40 electrons per Å$^2$ on Titan Krios (Thermo Fisher Scientific) at 300 kV using a Falcon 4i direct electron detector. Data collection was performed using the EPU v 3.0.0 data collection software (Thermo Fisher).

## Cryo-EM imaging

The data for *Tv*20S complexed with MZB was processed using cryoSPARC (v4.0.1). Images were imported and corrected using default settings for patch motion and patch contrast transfer function (CTF). Particles were selected using a blob picker (with a particle search size of 190 Å), extracted with a box size of 440 pixels, and classified using a 2D classification job. For *Tv*20S-MZB initial 100 2D classes were sorted using 2D classification. Particles and exposures were reduced to 6,135 micrographs by manual curation. Subsequent 2D classification revealed that of the 916,422 good particles, only 2% of the 2D classes contained the entire proteasome. After several rounds of 2D classification to remove unwanted particles, the subset of 13,933 particles (side views only) was used for an ab initio reconstruction and further homogeneous refinement. The resolution changed from 2.79 Å to 2.86 Å during 3D classification and removal of poor 3D classes and further reiterations including 2D classifications. We have observed rather a large gap between the unmasked-calculated resolution (3.73 Å with the 0.143 threshold and 6.91 Å with the 0.5 threshold). The C2 symmetry was applied in 3D homologous refinement steps for the generation of the final model. The cryo-EM map (GSFSC) achieved a final resolution of 2.86 Å, utilizing 14,257 particles.

The cryo-EM 3D map for *Tv*20S complexed with CP-17 was generated in a similar manner. Images were imported in small batches and then pooled, motion-corrected, and CTF-corrected in four batches, each containing approximately 2000 images. Particle picking was performed using well-defined *Tv*20S-MZB particles as templates. 2D classifications were also conducted in four batches, and subsequently, visually well-defined particles were selected (528,679), reclassified, and filtered using both 2D and 3D classification methods. After several rounds of further selection processes, 80,145 particles were retained for the final 3D map generation. The resulting cryo-EM map of *Tv*20S complexed with CP-17 exhibited a final resolution estimate of 2.60 Å, as determined by GSFSC analysis. An overview of cryo-EM data processing workflow is shown in Supplementary Fig. 10.

## Model building and refinement

A high-resolution proteasome structure of *Leishmania tarentolae* (PDB ID: 7ZYJ) was used as an initial template to build the model[47]. The initial model was built by alignment of *Tv*20S the structures predicted by AlphaFold onto the 7ZYJ template. The *Tv*20S-MZB model was fitted to density using ChimeraX[51] and refined by rigid body refinement and real space refinement in Coot 0.9.8.5 EL[52] and structure optimization using

ISOLDE package[53] in ChimeraX[54]. Representative images of the structures and maps were generated using ChimeraX and Pymol. The *Tv*20S-CP-17 structure was refined in a similar manner as the *Tv*20S-MZB structure using the *Tv*20S-MZB complex as a starting model. Local resolution maps for both cryoEM maps were estimated using individual half maps and conducting calculations in Phenix v1.20.1-4487[55]. The data were implemented into *Tv*20S-MZB and *Tv*20S-CP-17 maps and colored according to the local resolution estimates in ChimeraX[54] (Supplementary Fig. 11). The reported maximum resolution (Supplementary Table 2) for deposited Cryo-EM maps was determined through FSC analysis by applying a tight mask to selected Cryo-EM map, with a threshold of 0.143. The *Tv*20S-MZB structure was deposited to the world protein data bank under the ID number PDB ID: 8OIX and the *Tv*20S-CP-17 structure under PDB ID: 8P0T (Supplementary Table 2).

## Reporting summary

Further information on research design is available in the Nature Portfolio Reporting Summary linked to this article.

## Data availability

The atomic coordinates were deposited in the Protein Data Bank (https://www.rcsb.org) under the PDB accession codes 8OIX and 8P0T. Cryo-EM density maps are available under accession codes EMD-16901 and EMD 17337. Mass spectrometry data is available under accession code PXD051584. Links to the atomic coordinates used in this study: 4R3O, 5LF3, 7AWE, 7PG9, 7ZYJ, 8OIX, 8P0T. Source data are provided as a Source Data files. The proteomics data generated in this study are provided as Supplementary Data 1. These raw data are available at the following repositories: FTP link: ftp://massive.ucsd.edu/v07/MSV000094569/ or ProteomeXchange: PXD051584. Source data are provided with this paper.

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

## Acknowledgements

The data was collected at the Umeå Center for Electron Microscopy, a node of the Cryo-EM Swedish National Facility, funded by the Knut and Alice Wallenberg, Family Erling Persson and Kempe Foundations, Sci-LifeLab, Stockholm University and Umeå University. and we thank in particular to Michael Hall, Camilla Holmlund, and Linda Sandblad. We are grateful to Karim Rafie and Lar-Anders Carlson for invaluable help with cryo-EM experiments and data processing. The research was supported by NIH awards AI158612 and AI146387 to AJO, LE, and WHG, and DK120515 to LE. We would like to thank Dr. Samuel A. Myers, La Jolla Institute for Immunology for mass spectrometry support. PF received funding from the European Union's Horizon 2020 research and innovation program under the Marie Skłodowska-Curie grant agreement No [846688]. This research was funded by the project the National Institute Virology and Bacteriology (Program EXCELES, Project No. LX22NPO5103) - Funded by the European Union - Next Generation EU (awarded E.B.). Academy of Sciences of the Czech Republic, RVO: 61388963, is also acknowledged. BMH was supported in part by the UCSD Graduate Training Program in Cellular and Molecular Pharmacology through an institutional training grant from the National Institute of General Medical Sciences, T32 GM007752. PF would like to acknowledge Milan Fabry for providing advice on cloning and Martin Horn for assistance with enzymatic assays. JA would like to acknowledge the St. Baldrick's Foundation for the International Scholar award 2022–2025 and the deanship of scientific research at the University of Jordan for the scientific leave.

## Author contributions

J.S. collected and processed cryo-EM data, J.B., P.F., B.M.H. performed expression and enzyme experiments. Y.M. and L.E. cultured *T. vaginalis* and provided cell lysates, J.A. and W.H.G. designed and synthesized CP-17, P.F., A.J.O. and E.B. conceived the project. J.S., P.F., A.J.O. and E.B. wrote the manuscript. All authors critically reviewed the paper. A.J.O. and E.B. supervised the project. P.F., L.E., W.H.G., A.J.O., and E.B. obtained funding.

## Competing interests

The authors declare no competing interests.
