## [Peer Review File · Nature Communications]

Structural elucidation of recombinant *Trichomonas vaginalis* 20S proteasome bound to covalent inhibitorsReviewers' comments:

Reviewer #1 (Remarks to the Author):

The authors report the first recombinant overexpression of the *Trichomonas vaginalis* 20S proteasome (Tv20S) in the insect cells, which will facilitate many studies in the field. They also report cryo-EM structures of Tv20S bound to two inhibitors, revealing potential strategy for the development of species selective inhibitors in the future. Overall, this is a potentially useful study. However, there are significant concerns about the recombinant system they have developed, as well as major concerns on the data quality based on the PDB validation report and listed statistics in the SI Table.

Major concerns:

1. The authors added a C-terminal twin-strep tag to $\beta 7$ and expected this to have minimal effect on the assembly of the proteasome complex. However, previous studies have shown that $\beta 7$ is the last subunit incorporated into the half proteasome and the C-terminal tail is important for the holo-proteasome assembly and $\beta 1$ propeptide processing (Marques AJ, Glanemann C, Ramos PC, Dohmen RJ. The C-terminal extension of the beta7 subunit and activator complexes stabilize nascent 20 S proteasomes and promote their maturation. *J. Biol. Chem.* 2007; 282:34869–34876). I don't know about the Tv20S, but in other proteasomes, the $\beta 7$ C-terminal tail inserts into the space between $\beta 1$ and $\beta 2$ of the opposing half proteasome, and this interaction is considered the first event when two half-proteasomes come together. If this holds true in the Tv20S assembly, the C-terminal twin-strep tag may have hindered the assembly of the full complex. This would explain why there were so many immature half proteasome particles in their cryo-EM images, and the use of very low percentage of raw particles in 3D reconstruction.
2. Fig. 1b is confusing. Because *Trichomonas* doesn't contain the PAC1–PAC2 and PAC3–PAC4 chaperones, Tv likely uses PAC1-2 and PAC3-4 of the insect cells to help assemble the rTv20S, and this should be clearly stated. If so, Tv may also use the insect Ump-1, rendering the coexpression of Tv Ump-1 redundant? This can be easily figured out by comparing the Tv20S expression with or without the Ump-1 co-infection. A related question is the author's use of Ump-1 codon-optimization for *E. coli* in the SF9 cells. Are they sure the introduced Ump-1 is functional? In the introduction, the authors state the rTv20S is functional by biochemical comparison with native Tv20S. However, we didn't find any activity comparison between the native and rTv20S. It's important to perform a detailed comparison of the activity and stability between native and rTv20S.
3. In Fig. 1e, the native gel shows that the rTv20S runs higher than the native one. But it is hard for imagine a minor difference of a twin Strep tag (6kDa) could cause the observed shift in the native gel. Is it possible that the recombinant proteins have acquired additional modifications in the SF9 expression system? The authors should run an SDS-PAGE gel and perform mass spectrometry to find out.

4. it seems that the authors used merely 2% of total particles for final reconstruction. This raises the serious concern that vast majority of their rTv20S particle may not have matured or fully assembled. Could this be due to the Ump-1 codon not being optimized for SF9 expression, or the system has overwhelmed the limited supply of the native insect PAC1/2 and PAC3/4 chaperones? Or the purification tag may have interfered with the assembly? This should be investigated.

5. As 98% of the rTv20s become half proteasome after adding inhibitors (Fig. S2a), the stability of rTv20S is a concern. Will these inhibitors also destroy the wild type Ts20S? If the answer is yes, the measured inhibitory activity may not reflect the real inhibitions of all sites because the inhibitors may get into the active site through the open ring of half proteasome. Furthermore, if the inhibitors can break up most of the proteasome particles, the inhibition activity may be irrelevant? If the inhibitors only break up the rTv20s but not the native 20S, the result in Fig. 2c does not represent the real inhibition of the native Tv20s. Will the inhibitor also break up the human proteasome core particle?

6. Fig. 2d shows CP-17 may enhance the beta-1 activity. The author reasoned that the beta-1 substrate being directed to the beta-1 site when both beta-2 and beta-5 are blocked by CP-17. But this is confusing: if the substrate is only specific for beta-1 site, it won't go to the beta-2 and beta-5 sites. And blocking beta-2 and -5 should not affect the beta-1 activity. The author should use other beta-1 specific substrates to test the hypothesis.

7. Both structures describe only the residues of binding pockets. No potential interactions were proposed. Is there any polar interaction between inhibitors and the binding pockets? The author didn't mention any. Both inhibitors contain peptide bond structure which should contribute several polar interactions. Thr1 covalent bond and the π - π interaction may not be all interactions. The authors should do more detailed analysis.

8. The authors didn't provide the human PDB code(s) used in the selectivity analysis (Figs. 4-5), making it hard to trace the structural differences. We downloaded two human constitutive proteasome PDBs with or without ligand (5LF3 and 4R3O) and one human immunoproteasome with a ligand (7AWE). In both constitutive and immune-proteasome structures, we were unable to identify the described residues Ser122, Cys128, and Met130 in beta-2, nor Ser129 in beta-5. In fact, Ser123, Cys129, and Met131 of beta-3 and Ser129 of b6 might correspond to the residues the authors referred to. This makes us wonder if the structural analysis was done carefully, and whether chain ID and residue number were assigned incorrectly.

9. Using ligand free human 20S to compare with ligand-bound rTv20S is not convincing. The author stated the Met130 of beta-2 (assuming the residue # is correct) and Met45 of beta-5 will clash with CP-17. In fact, comparison of the ligand-bound with ligand-free human 20S (4R3O) show that Met45 of ligand-bound 20S (5LF3 and 7AWE) adopts a new rotamer to accommodate the ligand. Met has 13 rotamers. The author should consider solving the structure of human 20S with CP-17.

10. In Section 6 of validation report of 8OIX, the calculated raw map from half maps is barely visible in all projections. The resolution of calculated map at FSC=0.5 is 6.91 which is much worse than

the stated 3.31 Å. This may indicate inconsistency of the two half maps. Either the map quality should be improved, or the noted issue resolved.

11. From Fig. S8, MZB is most potent to b5. However, no inhibitor density was found in b5. Is this due to map quality or sample quality? The author should resolve this issue.

Minor concerns:

1. The submitted manuscript may not be the final version, because several communications between co-authors were not removed.

2. Insect expression of rTv20S is an important part of the manuscript. The description in Method is too brief, expression conditions e.g., expression time should be stated. In Fig. S1, the structures of the plasmids should be depicted clearly since seven genes are in one plasmid. Only sequences in plain text without any labeling, the construct structure is not clear. The polyhedrin promoter, Kozak sequence, start and stop sites of a gene, accession code of the gene, SV40 polyA tail should all be clearly labeled.

3. Production of the rTv20s in SF9 cells used a similar approach as for the human 20S. The authors should cite the recombinant human 20S paper.

4. MV151 should be noted the Proteasome Activity Probe (it is named as Me4BodipyFL-Ahx3Leu3VS in R&D Systems, not MV151, are they the same thing?).

5. Fig. S1 is inconsistent. The left panels indicate a contiguous fragment of alpha-1 to alpha-7 (or beta-1 to beta-7) was inserted into the pACEBac1 vector, but the right panel clearly shows that each inserted subunit contains its own promoter region and termination region.

6. In the “Cryo-EM imaging” section, Fig. S2C says 13,933 particles were used to for ab initio reconstruction, but the final model has 14,257 particles. It appears the 13,933 particles are side views, and they added 4,381 particles in top views to obtain the final 3D map. But it is not clear where and how they obtained these extra top views.

7. From the validation report (8OIX), it looks like the tag was removed and the beta5 propeptide was not processed. Is this true?

8. Fig 2c should include a Coomassie staining of the same gel to serve as a loading control (without fluorescence).

9. “CP-17, which inhibits the Tv20S β5 subunit with a 10-fold higher potency than the equivalent subunit of the human constitutive proteasome”: it is only 5-fold difference in ref 8?

10. What is homologous refinement in “The C2 symmetry was applied in 3D “homologous refinement” steps for generation of final model”. Is this a typo or using a homologous map as

reference? Please clarify.

11. In the Fig. S2c flow chart, the resolution of rTv20S-mzb was reduced from 2.70 Å to 2.86 Å. Similarly, the resolution of Tv20S-CP17 was reduced from 2.48 Å to 2.60 Å. Is there any reason to reduce the resolution? Also, the particle number was reduced. Did the authors use any procedures that were not included in the flow chart to reduce the particle number and the resolution?

12. The symmetry used should be included in the table. The clash scores (24.83, 28.72) and rotamer (2.41, 3.84, reported, 5.1 and 4.0- validation reports) are way too high for a 2.6 Å structure.

13. The RMS deviations of bond lengths and angles, b-factor of model and ligand, model composition, model resolution are all missing in the data statistics.

14. Local resolution map and model-map correlation are both missing.

15. Distance of clash should be labeled.

Reviewer #2 (Remarks to the Author):

This manuscript describes structures of *Trichomonas vaginalis* (Tv) 20S proteasome covalently bound to two small molecule proteasome inhibitors. Proteasome has been indicated as a therapeutic target for STDs caused by this protozoan parasite. Therefore, these results have value in the development of drugs to treat this widespread disease.

A significant technical advance reported in the paper is the production of recombinant Tv 20S proteasome. This technically challenging achievement overcomes previous problems in producing endogenous Tv proteasome in sufficient quantity and quality for biochemical and structural studies. The results show that the purified recombinant proteasome has features that are indistinguishable from endogenous Tv and therefore make the current work possible. In addition to the production and functional characterization of Tv 20S, this manuscript describes high resolution cryo electron microscopic structures of Tv20S with bound to previously identified Tv 20 inhibitors MZB and CP-17. The results show that whereas MZB binds to each of three different proteasome catalytic sites, CP-17 binds more selectively to only two: the beta 2 and beta 5 subunits. Overall, these results provide a platform for the possible future development of more specific Tv20S proteasome inhibitors as useful drugs.

Critique:

The manuscript is clearly written, and the results are largely convincing. I am not competent to critically judge all technical aspects of the structural determination and model building. Nevertheless, the overall description of the models seems biochemically reasonable and generally conform to the functional data. Likewise differences of inhibitors between Tv and human proteasomes can be explained by the structure reported here and elsewhere (e.g. the greater

specificity of CP-17 for Tv versus human 20S). This work builds on previous work on this topic this and other groups. The extent to which this represents a large breakthrough in the stated goal of developing new drugs is less certain. This is a step forward (aided by the technical advance) but not yet delivered and might be viewed differently by different people.

The data in Figure 2d should be expanded, in my opinion. In addition to a representative time course for a single inhibitor concentration, as shown, it would be useful to determine relative IC50s for these inhibitors with each of the subunit-specific substrates for both the recombinant and native enzymes. This would validate the broader claims about both relative specificity of the inhibitors (including differential sites) and the quality of the recombinant protein. This is an issue since the data show that only a small fraction of expressed protein fully assembles into intact 20S. This raises questions about interpretations of minor differences in structure. Finally, activities should be reported in more quantitative fashion as nmol AMC/min/ug. This will allow direct quantitative comparisons that are central to the conclusions.

Minor:

Pg 10, paragraph 2, line 1: “catalytical” should be “catalytic”.

Reviewer #3 (Remarks to the Author):

The manuscript from Boura and coworkers reports the structure of a fully recombinant 20S proteasome from the human pathogen *Trichomonas vaginalis* that is responsible for urinary tract infections. The manuscript reports structures of Tv20S bound to two different inhibitors at 2.8 and 2.6 Angstrom resolutions. While the technological advance of producing the Tv20S proteasome by separately expressing the alpha and beta subunits in Sf9 insect cells is noteworthy, overall the manuscript fails to significantly advance our understanding of TV20S function or its inhibition. For example, high resolution structures already exist for several bacterial proteasomes, including *Leishmania tarentolae* and *Mycobacterium tuberculosis* proteasomes and cryo-EM proteasome structures are no longer considered novel per se. Meanwhile, the inhibitor MZB was not observed in the beta5 subunit structure, despite its known ability to inhibit the proteolytic activity of this subunit, indicating incomplete applicability of the current technique toward visualizing covalent adducts of key beta subunit residues. Finally, while the authors hypothesize several opportunities to improve selective inhibitor design based on their structural conclusions regarding differences between human and Tv proteasomes; no evidence is provided that this can actually be accomplished with molecules based on the authors' comments of significant differences in shape of the active sites. Indeed, this exact challenge of selectivity has been explored extensively for Tuberculosis proteasome inhibitors and not led to any known TB therapeutics. Thus, the manuscript falls well short of providing any new leads based on the reported structure. Along these lines, it is already known that CP-17 inhibits Tv20S with 10-fold higher potency as the authors mention, which would have been a great starting point for optimization of the lead based on their structure. Finally, the conclusion that beta2 and beta5 inhibition by the inhibitor CP17 led to

increased activity at beta1 due to increased substrate concentration is obvious as the substrate is not divided among subunit "enzymes". However, it is unclear what the ramifications of this observation are, or indeed if the activity is the same at 1/3 total substrate with inhibited beta2 and beta5, as it would be for total substrate with no inhibition. The authors also state they performed enzyme kinetics, however, no k_{cat} and K_m values are reported and only gain in AMC fluorescence as a function of time is presented in Figure 2d. As an additional minor note, the manuscript appears hastily written because several personal comments from the authors have been left behind in the manuscript. With the overall scientific limitations in mind, this reviewer does not feel that the manuscript rises to the level of publication in Nature Communications. In its current form it is likely more appropriate for a biological structure-focused journal where the protein expression technique would be better appreciated.

Responses to Reviewers' comments:

Reviewer #1 (Remarks to the Author):

The authors report the first recombinant overexpression of the *Trichomonas vaginalis* 20S proteasome (Tv20S) in the insect cells, which will facilitate many studies in the field. They also report cryo-EM structures of Tv20S bound to two inhibitors, revealing potential strategy for the development of species selective inhibitors in the future. Overall, this is a potentially useful study. However, there are significant concerns about the recombinant system they have developed, as well as major concerns on the data quality based on the PDB validation report and listed statistics in the SI Table.

We appreciate that Reviewer #1 considers our study potentially useful and believes it will facilitate many studies in the field. We are grateful for all the suggestions by Reviewer #1 that we have now addressed and believe that these changes have improved our manuscript.

Major concerns:

1. The authors added a C-terminal twin-strep tag to $\beta 7$ and expected this to have minimal effect on the assembly of the proteasome complex. However, previous studies have shown that $\beta 7$ is the last subunit incorporated into the half proteasome and the C-terminal tail is important for the holo-proteasome assembly and $\beta 1$ propeptide processing (Marques AJ, Glanemann C, Ramos PC, Dohmen RJ. The C-terminal extension of the $\beta 7$ subunit and activator complexes stabilize nascent 20 S proteasomes and promote their maturation. *J. Biol. Chem.* 2007; 282:34869–34876). I don't know about the Tv20S, but in other proteasomes, the $\beta 7$ C-terminal tail inserts into the space between $\beta 1$ and $\beta 2$ of the opposing half proteasome, and this interaction is considered the first event when two half-proteasomes come together. If this holds true in the Tv20S assembly, the C-terminal twin-strep tag may have hindered the assembly of the full complex. This would explain why there were so many immature half proteasome particles in their cryo-EM images, and the use of very low percentage of raw particles in 3D reconstruction.

Response: There are numerous reasons why there is an abundance of immature half proteasomes. One hypothesis for this may be due to the absence of the *T. vaginalis* PAC1-PAC2 and PAC3-PAC4 assembly chaperones. It is likely that insect chaperones may be able to assist with the full proteasome assembly but are likely to be sub-optimal. It is difficult to test this hypothesis since we cannot find PAC1-PAC2 and PAC3-PAC4 equivalents in the *T. vaginalis* genome.

An alternative hypothesis is that the C-terminal tag on $\beta 7$ may slow down the assembly of the full 20S proteasome. To test this hypothesis, the recombinant *Tv* proteasome was enriched on the Streptactin XP column, and the eluted protein was incubated for 24 and 72 h at room temperature and compared to a sample that was stored at 4°C. All samples were run on a native gel after 72 h incubation, and we show that the amount of half-proteasome decreases with time following incubation at room temperature (SI Fig. 3c). These data reveal that a higher percentage of mature 20S proteasome can be achieved but requires additional time for the half-proteasomes to interact.

In the structures of the 20S proteasome our study revealed that the C-terminus of $\beta 7$ is correctly inserted into the space between $\beta 1$ and $\beta 2$ of the opposing half proteasome (new SI Fig. 7). In the resubmitted manuscript, we now discuss the C-terminus of $\beta 7$, and we also cite the study by Marques et al.

Specifically, we now state: "...by the observation of fully assembled *Tv*20S with the C-terminus of the $\beta 7$ subunit, that is known to stabilize nascent 20S (Marques et al.), properly inserted into the space between $\beta 1$ and $\beta 2$ of the opposing half proteasome (SI Fig. S7)."

2. Fig. 1b is confusing. Because *Trichomonas* doesn't contain the PAC1–PAC2 and PAC3–PAC4 chaperones, *Tv* likely uses PAC1-2 and PAC3-4 of the insect cells to help assemble the rTv20S, and this should be clearly stated. If so, *Tv* may also use the insect Ump-1, rendering the coexpression of *Tv* Ump-1 redundant? This can be easily figured out by comparing the *Tv*20S expression with or without the Ump-1 co-infection. A related question is the author's use of Ump-1 codon-optimization for *E. coli* in the SF9 cells. Are they sure the introduced Ump-1 is functional?

Response: We used *E. coli* codon optimized genes because *E. coli* prefers to use codons that have moderate GC content (high GC content codons are rare in the *E. coli* genome) and therefore these genes express well in every other common expression system including insect cells.

We have performed the suggested experiment where *Tv*20S was expressed with and without *Tv* Ump-1 and can conclusively show that the recombinant *Tv* Ump-1 is functional. When using the activity-based probe, Me4BodipyFL-Ahx3Leu3VS, Sf9 cell extracts contain a mixture of two functional proteasomes on a native gel, the Sf9 insect proteasome with a mw of ~720 kDa and the recombinant *Tv*20S proteasome with a mw of ~700 kDa. When the same probe-labelled samples are evaluated on a denaturing gel, there is very little labelling of the *Tv*20S β 5 subunit in the absence of *Tv*Ump-1 (compare Lane 2 vs Lane 3 in SI Fig. S3a). Therefore, *Tv* Ump-1 promotes full maturation of the recombinant *Tv*20S thereby revealing that it is functional.

We have added the following text to the results section of the manuscript: “To understand the role of *Tv* Ump-1, we expressed r*Tv*20S with and without this chaperone protein. In the absence of Ump-1, the eluted fraction from the streptavidin column contained more half-proteasomes, indicating incomplete assembly. Interestingly, incubating these proteins for 72 hours at room temperature partially rescued this defect (SI Fig. S3c). Additionally, β 5 subunit activity, measured using both the fluorogenic probe and substrate, was significantly weaker when Ump-1 was absent (SI Fig. S3d). These findings collectively demonstrate that recombinant *Tv*Ump-1 functions as a critical chaperone, essential for the complete maturation and full activity of r*Tv*20S”.

In the introduction, the authors state the r*Tv*20S is functional by biochemical comparison with native *Tv*20S. However, we didn't find any activity comparison between the native and r*Tv*20S. It's important to perform a detailed comparison of the activity and stability between native and r*Tv*20S.

Response: We have now performed an in-depth comparison of the native and recombinant enzymes and generated Michaelis Menton plots for both enzymes using the three subunit-selective substrates. A new Fig. 2c shows that the enzymes cleave the substrates at the same rate across a wide range of substrate concentrations. A table containing the K_M values was also generated and included as Table S1.

In addition, we pre-incubated n*Tv*20S and r*Tv*20S with 50 μ M of MZB or CP-17 for 1 h and then quantified the remaining activity with the fluorogenic substrates. MZB and CP-17 had similar inhibitory effects on proteasome activity (Fig. 2d). MZB inhibited the β 1 activity of n*Tv*20S and r*Tv*20S to the same level, however CP-17 activated n*Tv*20S by 200% while the same compound activated r*Tv*20S by only 120%. It is unclear why the native β 1 subunit activates more than the recombinant enzyme. As outlined in the manuscript, we do not see any structural differences in the β 1 subunit in the presence of either MZB and CP-17 and therefore the general mechanism of subunit activation cannot be explained based on structure. We hypothesize that the local concentration of β 1 substrate is increased when one or more subunits is inhibited.

3. In Fig. 1e, the native gel shows that the r*Tv*20S runs higher than the native one. But it is hard to imagine a minor difference of a twin Strep tag (6kDa) could cause the observed shift in the native gel. Is it possible that the recombinant proteins have acquired additional modifications in the Sf9 expression system? The authors should run an SDS-PAGE gel and perform mass spectrometry to find out.

Response: We have performed an in-depth comparison of the two protein complexes and have been unable to find modifications to the recombinant enzyme that would cause the observed mass difference seen on the native gels. The following studies were performed.

1. r*Tv*20S and n*Tv*20S were incubated with the Me4BodipyFL-Ahx3Leu3VS probe and run on a denaturing gel. The probe strongly labelled the β 2 (upper) and β 5 (lower) bands on the gel and weakly labeled the β 1 band (middle). When comparing the first and fourth lanes, the banding pattern for r*Tv*20S and n*Tv*20S are identical. This gel was then silver stained and all of the subunits bands can now be seen. While the r*Tv*20S preparation is much cleaner than n*Tv*20S the banding pattern in the region of 25 kDa to 30 kDa is the same. Therefore, from these studies, there is no evidence that the r*Tv*20S have higher molecular mass than the n*Tv*20S subunits.
2. The intact insect proteasome is clearly higher in molecular mass than n*Tv*20S and therefore, we hypothesized that some of the insect proteasome subunits may get incorporated into

rTv20S during the assembly. This would then increase the overall mass of rTv20S. We therefore performed a proteomic analysis on the recombinant enzyme and searched the resulting data against the *T. vaginalis* and *Spodoptera frugiperda* proteomes. We found that the 14 most abundant subunits match to the *T. vaginalis* proteome (new Fig. 1g). Several insect proteasome subunits were found but their abundance was significantly lower (>100-fold). These studies indicate that population of rTv20S expressed in insect cell lines may contain a low percentage of host subunits mass but there was no dominant Tv20S/Sf20S chimeric proteasome present that could explain the slightly different masses between rTv20S and nTv20S.

4. it seems that the authors used merely 2% of total particles for final reconstruction. This raises the serious concern that vast majority of their rTv20S particle may not have matured or fully assembled. Could this be due to the Ump-1 codon not being optimized for SF9 expression, or the system has overwhelmed the limited supply of the native insect PAC1/2 and PAC3/4 chaperones? Or the purification tag may have interfered with the assembly? This should be investigated.

Response: We have already addressed concerns about the function of Ump-1, the chaperones and the potential interference by the purification tag. Our use of only ~2% of all the particles has no impact on the quality of the model of the fully assembled proteasome as only these particles were selected for reconstruction (Fig. S10).

5. As 98% of the rTv20s become half proteasome after adding inhibitors (Fig. S2a), the stability of rTv20S is a concern. Will these inhibitors also destroy the wild type Tv20S? If the answer is yes, the measured inhibitory activity may not reflect the real inhibitions of all sites because the inhibitors may get into the active site through the open ring of half proteasome. Furthermore, if the inhibitors can break up most of the proteasome particles, the inhibition activity may be irrelevant? If the inhibitors only break up the rTv20s but not the native 20S, the result in Fig. 2c does not represent the real inhibition of the native Tv20s. Will the inhibitor also break up the human proteasome core particle?

Response: The presence of half proteasomes is not a result of addition of inhibitor, as we observed half proteasomes during the purification process (in the absence of MZB).

6. Fig. 2d shows CP-17 may enhance the beta-1 activity. The author reasoned that the beta-1 substrate being directed to the beta-1 site when both beta-2 and beta-5 are blocked by CP-17. But this is confusing: if the substrate is only specific for beta-1 site, it won't go to the beta-2 and beta-5 sites. And blocking beta-2 and -5 should not affect the beta-1 activity. The author should use other beta-1 specific substrates to test the hypothesis.

Response: For substrates to be cleaved by a proteasome, they will ultimately sample all binding sites within the central core of the proteasome. Therefore, even though the β_1 substrate is not cleaved by the β_2 or β_5 subunit, it will interact with them, albeit for a very brief time. If one of the substrate binding pockets is blocked with an inhibitor, then the β_1 substrate will spend less time sampling this site and therefore the apparent concentration within the core of the proteasome would increase. In our Michaelis Menton plot (Fig 2c), we show that activity of Ac-RYFD-amc increases with increasing substrate concentration. Therefore, it is likely that the increase in β_1 activity (in the presence of CP-17) is also due to an increase in substrate concentration.

To examine this, rTv20S was pre-incubated with 5 μM of CP-17 to inhibit β_5 and then assayed with increasing concentrations of β_1 substrate. When compared to a control assay that lacks CP-17, there is between 120% to 160% increase in β_1 activity at each of the substrate concentrations tested.

Therefore, for example, when rTv20S is pre-treated with CP-17 and then assayed with 126 μM of Ac-RYFD, the velocity of the reaction is 1.4 RFU/sec (SI Fig. S4). To achieve this velocity in the non-inhibited assay, then ~200 μM of substrate is needed. Therefore, in the presence of CP-17 and 126 μM of Ac-RYFD-amc, the apparent concentration of substrate in the β_1 active site of Tv20S is equivalent to 200 μM in the non-inhibited enzyme.

In a separate study, we have shown that inhibition of nTv20S β_2 with leupeptin results in increased activity of β_1 while inhibition of β_1 and β_5 with ixazomib results in activation of β_2 . (Fajtova et al. Development of subunit selective

substrates for *Trichomonas vaginalis* proteasome. bioRxiv 2023). Furthermore, when $\beta 5$ is inhibited with either carfilzomib or CP-17 we see activation of $\beta 1$. Taken together these data reveal that an increase in the rate of substrate cleavage occurs when the substrate cannot sample other subunits within the core of the proteasome.

In the discussion section, we have added the following text “We propose that the higher activity is due to increased substrate concentration at the $\beta 1$ subunit. With $\beta 2$ and $\beta 5$ blocked by CP-17, the substrate has only $\beta 1$ available for interaction, leading to a higher observed activity compared to the uninhibited enzyme.”

7. Both structures describe only the residues of binding pockets. No potential interactions were proposed. Is there any polar interaction between inhibitors and the binding pockets? The author didn't mention any. Both inhibitors contain peptide bond structure which should contribute several polar interactions. Thr1 covalent bond and the π - π interaction may not be all interactions. The authors should do more detailed analysis.

Response: We acknowledge that our manuscript did not thoroughly discuss certain interactions. Our revised manuscript now contains three new figures (**Fig. S12 - S14**) that present an extensive and more detailed analysis, emphasizing the contributions of all relevant interactions between the inhibitors and the binding pockets.

8. The authors didn't provide the human PDB code(s) used in the selectivity analysis (Figs. 4-5), making it hard to trace the structural differences. We downloaded two human constitutive proteasome PDBs with or without ligand (5LF3 and 4R3O) and one human immunoproteasome with a ligand (7AWE). In both constitutive and immunoproteasome structures, we were unable to identify the described residues Ser122, Cys128, and Met130 in beta-2, nor Ser129 in beta-5. In fact, Ser123, Cys129, and Met131 of beta-3 and Ser129 of $\beta 6$ might correspond to the residues the authors referred to. This makes us wonder if the structural analysis was done carefully, and whether chain ID and residue number were assigned incorrectly.

Response: We apologize that PDB code for human 20S were omitted and the chain IDs for individual residues were not included. In the revised manuscript Fig. 4 and 5 have been corrected. We used one of the suggested human structures (**PDB ID = 4R3O**) and also included a new SI figure (**Fig. S15**) that illustrates the major structural differences between h20S vs Tv20S in the vicinity of the CP-17 inhibitor binding sites (beta-2 and beta-5).

9. Using ligand free human 20S to compare with ligand-bound rTv20S is not convincing. The author stated the Met130 of beta-2 (assuming the residue # is correct) and Met45 of beta-5 will clash with CP-17. In fact, comparison of the ligand-bound with ligand-free human 20S (4R3O) show that Met45 of ligand-bound 20S (5LF3 and 7AWE) adopts a new rotamer to accommodate the ligand. Met has 13 rotamers. The author should consider solving the structure of human 20S with CP-17.

Response: Thank you for this suggestion. The structural superposition has now been done as suggested by Reviewer #1. Specifically, we have carefully re-examined the structural analysis/alignments with all three suggested structures (4R3O, 5LF3 and 7AWE) in addition to 7PG9, which we have used previously. We have rephrased our statements about differences between human 20S proteasome and Tv20S. In the discussion we now more carefully state the possibilities of exploiting these differences in the future research and drug development.

Thank you for the suggestion to solve the structure of the human proteasome in complex with CP-17. While this is beyond the scope of the current study, the recombinant expression of the human proteasome and structural studies with peptide epoxyketone inhibitors are a long-term goal of this project.

10. In Section 6 of validation report of 8OIX, the calculated raw map from half maps is barely visible in all projections. The resolution of calculated map at FSC=0.5 is 6.91 which is much worse than the stated 3.31 Å. This may indicate inconsistency of the two half maps. Either the map quality should be improved, or the noted issue resolved.

Response: We apologize for a lack of clarity in the validation report, but we do not have control over the quality of the maps' contours in the PDB report. Here, we present a figure of raw maps with different contours solely for the reviewer's revision (Figure for Reviewer 1).

We would like to clarify the other concern. The aforementioned resolution is an estimate of Fourier shell correlation curve at 50% FSC(0.5) = 6.9. This corresponds to a resolution estimate calculated for the RAW map at FSC 0.5. The author-provided FSC(0.5)=3.31 which corresponds to resolution of FCS curve calculated for a tight mask applied on the map. Moreover, the maximum resolution reported in the manuscript (e.g., [8OIX] = 2.86 Å) is determined through FSC analysis, employing a specific threshold of 0.143.

Now we state in the manuscript (methods section): “The reported maximum resolution (Table S3) for deposited Cryo-EM maps was determined through FSC analysis by applying a tight mask to selected Cryo-EM map, with a threshold of 0.143.”

11. From Fig. S8, MZB is most potent to b5. However, no inhibitor density was found in b5. Is this due to map quality or sample quality? The author should resolve this issue.

Response: The map of the $\beta 5$ active site does not allow to model a bound MZB molecule (Fig. S6c). It is highly improbable that the absence of a clear electron density is attributable to poor sample quality, as all the residues of the $\beta 5$ active site are well resolved. Our biochemical data (Fig. 2) confirms that the inhibitor is bound. The lack of density, therefore, is probably caused by the flexibility of the inhibitor in the $\beta 5$ active site.

We now state in the manuscript: “MZB was found to bind to six sites within Tv20S that corresponded to three subunits in each β -ring, albeit the electron density was insufficient to fully resolve its binding to the $\beta 5$ subunit.”

Minor concerns:

1. The submitted manuscript may not be the final version, because several communications between co-authors were not removed.

Response: We apologize for this oversight. It has been corrected.

2. Insect expression of rTv20S is an important part of the manuscript. The description in Method is too brief, expression conditions e.g., expression time should be stated. In Fig. S1, the structures of the plasmids should be depicted clearly since seven genes are in one plasmid. Only sequences in plain text without any labeling, the construct structure is not clear. The polyhedrin promoter, Kozak sequence, start and stop sites of a gene, accession code of the gene, SV40 polyA tail should all be clearly labeled.

Response: Thank you for the suggestion. The SI Figure (currently Fig. S2) has been modified as suggested, and indeed, it is now clearer.

3. Production of the rTv20s in SF9 cells used a similar approach as for the human 20S. The authors should cite the recombinant human 20S paper.

Response: We do cite the study by Toste Rego, A. & da Fonseca (Characterization of Fully Recombinant Human 20S and 20S-PA200 Proteasome Complexes). We are not aware of other studies that have expressed recombinant human 20S that should be cited.

4. MV151 should be noted the Proteasome Activity Probe (it is named as Me4BodipyFL-Ahx3Leu3VS in R&D Systems, not MV151, are they the same thing?).

Response: Thank you for asking us this question. We assumed that MV151 and Me4BodipyFL-Ahx3Leu3VS were the same compound, but they are actually slightly different. MV151 is BodipyTMR-Ahx3Leu3VS and therefore has a tetramethylrhodamine fluorescent label while Me4BodipyFL-Ahx3Leu3VS contains a fluorescein dye. We have updated the manuscript and used Me4BodipyFL-Ahx3Leu3VS instead of MV151.

5. Fig. S1 is inconsistent. The left panels indicate a contiguous fragment of alpha-1 to alpha-7 (or beta-1 to beta-7) was inserted into the pACEBac1 vector, but the right panel clearly shows that each inserted subunit contains its own promoter region and termination region.

Response: Each subunit has its own promoter region and termination region. These were also highlighted in the updated Fig. S2.

6. In the “Cryo-EM imaging” section, Fig. S2C says 13,933 particles were used to for ab initio reconstruction, but the final model has 14,257 particles. It appears the 13,933 particles are side views, and they added 4,381 particles in top views to obtain the final 3D map. But it is not clear where and how they obtained these extra top views.

Response: We apologize for the confusion; 13,724 particles were used for the *ab initio* model. In the next step, 13,933 side view particles were used to obtain an improved model using homogeneous reconstruction. Then 4381 top view particles were added in next round. For the final model, more particles were removed during the several rounds of homogeneous reconstruction and 3D classifications. Please note: clear top views of full proteasome were assumed to be the particles with highest contrast. Figure S10 and its legend was modified to improve the clarity of the processing.

We now state in the Figure legend:

Fig. S10. Cryo-EM workflow of data processing. This image processing workflow was employed to reconstruct the Tv20S structures in cryo-EM. Both datasets were acquired at Titan Krios with the Falcon 4i detector under identical conditions using the same setup (refer to Supplementary Table 1 for details). The ab initio model served as the starting point for 3D homogeneous refinement (Homogeneous Refinement) to enhance the quality of the maps. Multiple iterations, including 3D classification and 2D classification, were carried out in several rounds to eliminate unwanted particles, refine the resolution, and improve the maps. The unsharpened maps of the final reconstruction and the gold-standard Fourier Shell Correlation (FSC) curve using different masks are shown. The figures of the maps were generated by ChimeraX2. The resolution in italics corresponds to a particular Fourier Shell Correlation Cryosparc and was used to navigate the process of data analysis. The final resolution was estimated by a Fourier Shell Correlation job in Cryosparc when a tight mask was applied and was estimated to be 2.86 Å for 8IOX and 2.60 Å for 8P0T. The final resolution was estimated by Fourier Shell Correlation job in Cryosparc3 when a tight mask was applied and was estimated to be 2.86 Å for 8IOX and 2.60 Å for 8P0T. An example of the Tv20S-MZB route of processing is as follows: 1,436,978 particles were extracted from 6135 processed images. After several rounds of 2D classification to sort out unwanted classes, only 40,916 particles remained. These were then reclassified with independent 2D classifications for the top views and the side views. Separately, classes with 13,724 particles of both views were used for an ab initio model. This initial cryo-EM map was used as a starting model for homogeneous reconstruction, where 13,933 side view particles were used. Next, 4381 particles corresponding to the top views were added in the next round. For the final model, additional particles were removed during several rounds of homogeneous reconstruction and 3D classifications. Note: clear top views of the full proteasome were assumed to be the particles with the highest contrast.

7. From the validation report (8OIX), it looks like the tag was removed and the beta5 propeptide was not processed. Is this true?

Response: We apologize for this error. The entire sequence of the propeptide was included in the deposition. The deposition was updated not to contain the sequences of the propeptides.

8. Fig 2c should include a Coomassie staining of the same gel to serve as a loading control (without fluorescence).

Response: We added the same silver-stained gel after fluorescence visualization to Fig. 2b.

9. “CP-17, which inhibits the Tv20S β 5 subunit with a 10-fold higher potency than the equivalent subunit of the human constitutive proteasome”: it is only 5-fold difference in ref 8?

Response: Thank you for catching this. We have updated the manuscript now to indicate 5-fold.

10. What is homologous refinement in “The C2 symmetry was applied in 3D “homologous refinement” steps for generation of final model”. Is this a typo or using a homologous map as reference? Please clarify.

We apologize for misunderstanding; this was a typo. The corrected term is now the 3D homogeneous refinement.

11. In the Fig. S2c flow chart, the resolution of rTv20S-mzb was reduced from 2.70 Å to 2.86 Å. Similarly, the resolution of Tv20S-CP17 was reduced from 2.48 Å to 2.60 Å. Is there any reason to reduce the resolution? Also, the particle number was reduced. Did the authors use any procedures that were not included in the flow chart to reduce the particle number and the resolution?

Response: We apologize for a lack of clarity in this respect. We have now corrected the resolution. The resolutions that are only produced by a particular Fourier Shell Correlation Cryosparc run were used to navigate the process of 3D refinement. Final resolution is estimated by Fourier Shell Correlation job in Cryosparc when a tight mask was applied (detailed in Fig. S10). Several iterations of individual processing steps were often carried out. The final local resolution maps are now shown in new Fig. S11.

12. The symmetry used should be included in the table. The clash scores (24.83, 28.72) and rotamer (2.41, 3.84, reported, 5.1 and 4.0- validation reports) are way too high for a 2.6 Å structure.

Response: We have significantly improved our models clash score (11.17 & 11.9) and rotamers geometry (0.93 & 1.21), and redeposited the models into the PDB database. The new validation reports are attached. We have also included symmetry of the model (C2) in Table S3.

13. The RMS deviations of bond lengths and angles, b-factor of model and ligand, model composition, model resolution are all missing in the data statistics.

Response: The RMSDs of angles and lengths along with other details are now included in updated SI Table S3.

14. Local resolution map and model-map correlation are both missing.

Response: Both local resolution map and model-map correlation are now included as new Fig. S11 and Table S3.

15. Distance of clash should be labeled.

Response: It is unclear to us what exactly should be labeled. However, with some additional explanation for Reviewer #1, we will implement all the requested changes.

Reviewer #2 (Remarks to the Author):

This manuscript describes structures of *Trichomonas vaginalis* (Tv) 20S proteasome covalently bound to two small molecule proteasome inhibitors. Proteasome has been indicated as a therapeutic target for STDs caused by this protozoan parasite. Therefore, these results have value in the development of drugs to treat this widespread disease. A significant technical advance reported in the paper is the production of recombinant Tv 20S proteasome. This technically challenging achievement overcomes previous problems in producing endogenous Tv proteasome in sufficient quantity and quality for biochemical and structural studies. The results show that the purified recombinant proteasome has features that are indistinguishable from endogenous Tv and therefore make the current work possible. In addition to the production and functional characterization of Tv 20S, this manuscript describes high resolution cryo electron microscopic structures of Tv20S with bound to previously identified Tv 20 inhibitors MZB and CP-17. The results show that whereas MZB binds to each of three different proteasome catalytic sites, CP-17 binds more selectively to only two: the beta 2 and beta 5 subunits. Overall, these results provide a platform for the possible future development of more specific Tv20S proteasome inhibitors as useful drugs.

We appreciate that Reviewer #2 believes that our results have value in the development of drugs to treat this widespread disease and provide a platform for the possible future development of more specific Tv20S proteasome inhibitors as useful drugs.

Critique:

The manuscript is clearly written, and the results are largely convincing. I am not competent to critically judge all technical aspects of the structural determination and model building. Nevertheless, the overall description of the models seems biochemically reasonable and generally conform to the functional data. Likewise differences of inhibitors between Tv and human proteasomes can be explained by the structure reported here and elsewhere (e.g. the greater specificity of CP-17 for Tv versus human 20S). This work builds on previous work on this topic this this and other groups. The extent to which this represents a large breakthrough in the stated goal of developing new drugs is less certain. This is a step forward (aided by the technical advance) but not yet delivered and might be viewed differently by different people.

Response: For all successful protease inhibitor drug development programs, access to the recombinant enzyme is key for hit-to-lead optimization. Examples include the recombinant proteases from SARS-CoV-2, Hepatitis C Virus and HIV for antiviral drug development, recombinant renin and angiotensin-converting enzyme for anti-hypertensive drug development and recombinant dipeptidyl peptidase IV for diabetes drug development. One of the major bottlenecks for development of proteasome inhibitors for treatment of malaria (*Plasmodium falciparum*), leishmaniasis

(*Leishmania donovani*), Chagas disease (*Trypanosoma cruzi*) and African sleeping sickness (*Trypanosoma brucei*) has been the availability of a consistent source of enzyme for compound screening, hit-to-lead studies and structural studies. Hence, although the parasite proteasomes have been validated as a target for many years, there are only two proteasome inhibitors in late-stage clinical trials. Our lab aims to make recombinant proteasome for each of the above listed parasites however, our initial focus has been on *T. vaginalis*. We believe that the successful expression of a fully functional 28-subunit proteasome is a breakthrough moment for the field and know that other researchers believe so too. For example, when this manuscript was posted on bioRxiv, we were contacted by Novartis and have now successfully helped them to express a target proteasome for one of their internal anti-parasitic programs.

The data in Figure 2d should be expanded, in my opinion. In addition to a representative time course for a single inhibitor concentration, as shown, it would be useful to determine relative IC50s for these inhibitors with each of the subunit-specific substrates for both the recombinant and native enzymes. This would validate the broader claims about both relative specificity of the inhibitors (including differential sites) and the quality of the recombinant protein. This is an issue since the data show that only a small fraction of expressed protein fully assembles into intact 20S. This raises questions about interpretations of minor differences in structure. Finally, activities should be reported in more quantitative fashion as nmol AMC/min/ug. This will allow direct quantitative comparisons that are central to the conclusions.

Response: Thank you for this suggestion. We have now performed an in-depth comparison of the native and recombinant enzymes and generated Michaelis Menton plots for both enzymes using the three subunit-selective substrates. These data show that the two enzymes cleave the substrates at the same rate across a wide range of substrate concentrations as shown in the figure below. This figure is now included in the modified manuscript as Fig. 2c and the activities have been quantified as μmol of AMC/min/ μg . A table containing the k_{cat} and K_{M} values was also generated and included as Table S1.

We pre-incubated nTv20S and rTv20S with 50 μM of MZB or CP-17 for 1 hour and quantified the remaining activity using fluorogenic substrates. MZB and CP-17 showed similar inhibitory effects on proteasome activity, as shown in Fig. 2d of the revised manuscript. MZB inhibited $\beta 1$ activity equally in both nTv20S and rTv20S, while CP-17 activated nTv20S by 200% and rTv20S by 120%. The reason for the higher activation of the native $\beta 1$ subunit is unclear, as no structural differences were observed in the presence of MZB or CP-17. We propose that this increase is due to a higher substrate concentration within the proteasome core (see in-depth explanation above).

Minor:

Pg 10, paragraph 2, line 1: “catalytical” should be “catalytic”.

Response: Thank you for catching this typo. It has been changed.

Reviewer #3 (Remarks to the Author):

The manuscript from Boura and coworkers reports the structure of a fully recombinant 20S proteasome from the human pathogen *Trichomonas vaginalis* that is responsible for urinary tract infections. The manuscript reports structures of Tv20S bound to two different inhibitors at 2.8 and 2.6 Angstrom resolutions. While the technological advance of producing the Tv20S proteasome by separately expressing the alpha and beta subunits in Sf9 insect cells is noteworthy, overall the manuscript fails to significantly advance our understanding of TV20S function or its inhibition. For example, high resolution structures already exist for several bacterial proteasomes, including *Leishmania tarentolae* and *Mycobacterium tuberculosis* proteasomes and cryo-EM proteasome structures are no longer considered novel per se.

Response: In this study, the value of the cryo-EM structure was to understand the binding mode of two active site inhibitors. This would not have been possible with the native enzyme as sufficiently high yields and purity could not be achieved. As outlined above, access to a recombinant proteasome is essential for hit-to-lead development of other protease inhibitors that were subsequently approved as drugs. Having a structure will allow us to do structure-based drug design for the next generation of compounds. In addition, if resistant mutants arise, then understanding the structural changes of these mutations will be essential for hit-to-lead optimization. Therefore, we believe that the combination of the expression studies and structural studies add extensive novelty to this project. In the two examples

provided by Reviewer #3, it should be noted that the *Mycobacterium tuberculosis* proteasome consists of just one α -subunit and one β -subunit that each form into rings of seven. Therefore, this enzyme is simpler than the Tv20S proteasome reported here that contains seven different α -subunits and seven different β -subunits. Also, *Leishmania tarentolae* protozoa is used as an expression system for making recombinant proteasomes. While the cryo-EM structure of this compound has been published, it is a non-pathogenic organism and therefore important pathogenic proteasomes need to be modeled off of this structure. Our study is the first to solve the structure of a recombinant eukaryotic proteasome from a pathogenic organism that can then be directly used for structure-based drug design.

Meanwhile, the inhibitor MZB was not observed in the beta5 subunit structure, despite its known ability to inhibit the proteolytic activity of this subunit, indicating incomplete applicability of the current technique toward visualizing covalent adducts of key beta subunit residues.

Response: As stated above (Response to Referee #1), the map of the β 5 active site does not allow to model a bound MZB molecule (Fig. S6c). It is highly improbable that the absence of a clear electron density is attributable to poor sample quality, as all the residues of the β 5 active site are well resolved. Our biochemical data (Fig. 2) confirms that the inhibitor is bound. The lack of density, therefore, is probably caused by the flexibility of the inhibitor in the β 5 active site.

Finally, while the authors hypothesize several opportunities to improve selective inhibitor design based on their structural conclusions regarding differences between human and Tv proteasomes; no evidence is provided that this can actually be accomplished with molecules based on the authors' comments of significant differences in shape of the active sites. Indeed, this exact challenge of selectivity has been explored extensively for Tuberculosis proteasome inhibitors and not led to any known TB therapeutics. Thus, the manuscript falls well short of providing any new leads based on the reported structure. Along these lines, it is already known that CP-17 inhibits Tv20S with 10-fold higher potency as the authors mention, which would have been a great starting point for optimization of the lead based on their structure.

Response: This manuscript describes the recombinant expression and biochemical validation of Tv20S and the structure of the enzyme complex bound to two inhibitors. To perform structure-based drug design using CP-17 as a starting point, would require a chemistry team. These libraries will need to be iteratively tested and the cryo-EM structures solved. While the long-term goal of the project is to develop new leads, it is beyond the scope of this manuscript that new lead compounds are identified here. We have updated the discussion to suggest a key Cys residue in the β 1 subunit that could be targeted with a CP-17 analog.

Finally, the conclusion that beta2 and beta5 inhibition by the inhibitor CP17 led to increased activity at beta1 due to increased substrate concentration is obvious as the substrate is not divided among subunit "enzymes". However, it is unclear what the ramifications of this observation are, or indeed if the activity is the same at 1/3 total substrate with inhibited beta2 and beta5, as it would be for total substrate with no inhibition.

Response: Thank you for your question. We have addressed a similar question from Reviewers 1 and 2. Please see our answers above.

The authors also state they performed enzyme kinetics, however, no k_{cat} and K_M values are reported and only gain in AMC fluorescence as a function of time is presented in Figure 2d.

Response: Thank you for this suggestion. We have determined the K_M and k_{cat} values for each of the substrates using both nTv20S and rTv20S. The K_M plots are now included as Fig. 2c and the k_{cat} and K_M values are included in Table S1.

As an additional minor note, the manuscript appears hastily written because several personal comments from the authors have been left behind in the manuscript.

Response: We apologize for this oversight. It has been corrected.

With the overall scientific limitations in mind, this reviewer does not feel that the manuscript rises to the level of publication in Nature Communications. In its current form it is likely more appropriate for a biological structure-focused journal where the protein expression technique would be better appreciated.

Response: Through the valuable feedback provided by all Reviewers, we have substantially enhanced the quality of our manuscript. We are now confident that it meets the high standards required for publication in Nature Communications.

REVIEWER COMMENTS

Reviewer #1 (Remarks to the Author):

The authors have addressed many of our concerns. The revised manuscript now provides a better description and characterization of their expression system. A few additional comments are listed below.

1. In the Tv20S-MZB dataset (Fig. S10), only 2% of particles were fully assembled proteasomes while the remaining were half proteasomes. Please provide a full atlas of 2D classes in the first round which will better reflect the particle distribution of half and full proteasomes.
2. In the Tv20S-MZB dataset (Fig. S10), the particle number was reduced from 18,314 to 14,257 but the resolution was decreased from 2.79 to 2.86. But the authors state (inconsistently) in Method “cryo-EM imaging” section: The resolution was improved by 3D classification and removal of poor 3D classes and further reiterations including 2D classifications”.
3. In Method “cryo-EM imaging” section, particles and exposures were reduced to 6,135 particles by manual curation” – did the author mean micrographs rather than particles??
4. In the validation report of PDB ID 8P0T, the particle number was 197,322, the electron dose was 50 e/Å², and the defocus was 800-3600 nm. However, the particle number in Fig. S10 is different. The dose was 40 e/Å² and the defocus was 800-2400 nm in Table S3.
5. In the validation report of PDB ID 8OIX, the electron dose was 50 e/Å², and the defocus was 800-3600 nm. However, the dose was 40 e/Å² and the defocus was 800-2400 nm in Table S3.
6. In the validation report of PDB ID 8OIX, the Unmasked-calculated resolution is 3.73 Å with 0.143 threshold and 6.91 Å with the 0.5 threshold. This is an unusually large gap, indicative of low signal-to-noise ratio, likely due to limited number of particles (14,257) in the final reconstruction. The author should either note this shortcoming or collect more data to address the problem.
7. In Fig. S3c, Tv20S is more abundant in the -Ump1 samples, does not appear to agree with the description that “in the absence of Ump-1, the eluted fraction from the streptavidin column contained more half-proteasomes, indicating incomplete assembly”.
8. In Fig. S3b,d, why did the -Ump1 sample lose b5 binding to the fluorogenic probe? Does Ump-1 affect the b5 maturation?
9. Fig. 1e and Fig S3b: There is no corresponding band in the silver stain gel compared with the fluorogenic probe binding gel. Are they from the same gel?
10. Fig. 2c: the curves for the b1 of nTv20S and rTv20S never reached V_{max}, indicating low affinity or inaccessible to the b1 site. Consider another substrate?
11. From Fig 1c,f, the purified rTv20S was intact, so the proteasome was apparently destroyed by MZB, raising the question of the inhibitor entry into the active site. Several scenarios can be pictured, and different scenarios may result in different structures and kinetics. The author should address if half proteasome binds the inhibitor.
12. Fig. 4e-f and in page 9: there is no Fig 4f. Ile35 mentioned in the text but not labeled in the figure.

Reviewer #2 (Remarks to the Author):

This revised manuscript about structural relationships between two established proteasome inhibitors and the *Trichomonas vaginalis* 20S proteasome addresses many of the concerns raised by me and the other reviewers. In my view, a main advance in this manuscript is the description of a recombinant method of producing Tv proteasome for development of therapeutically useful inhibitors. That said, despite its usefulness for Tv, analogous methodology for other versions of proteasomes is now becoming more common and is therefore less novel. One recent paper not cited here (in their response the authors state that they are not aware.....) includes Adolf et al Nat Struc Mol Biol (<https://doi.org/101038/s41594-024-01268-9>); maybe this paper crossed during preparation of the current manuscript. The promise that the differences between the current 20S Tv structure and human 20S will lead to Tv-specific inhibitors remains to be determined. The description of structural features of Tv 20S with known inhibitors is useful new information.

One issue that I continue to find confusing (also raised by Reviewer 1) is the low percentage of fully assembled recombinant protein in the presence of inhibitor used for determination of the structure. That seems at odds with the apparent homogeneous preparations of purified protein on native gels. Is the inhibitor promoting dissociation? Why not compare Tv 20S +/- inhibitors on native gels? Did I miss something? On a related issue, the data in Figure S3c/d seem odd. The vast majority of the samples appears to be mature 20S.

Reviewer #3 (Remarks to the Author):

The authors have nicely rewritten the manuscript to highlight the new elements of their work and performed additional experimentation in response to reviewer comments. Prior to publication, the Km and kcat data should be reported with values and units, not just raw traces. While the overall conclusions are still somewhat modest, other than the demonstration that the proteasome can be purified from Sf9 cells, the rewritten manuscript does make a strong point of indicating that different inhibitors target different active sites in the proteasome. Therefore, I recommend publication of a suitably revised manuscript with Km and kcat values in tabulated format with error bars and number of repeats indicated.

REVIEWER COMMENTS

Reviewer #1 (Remarks to the Author):

The authors have addressed many of our concerns. The revised manuscript now provides a better description and characterization of their expression system. A few addition comments are listed below.

We are happy that the Expert Reviewer #1 believes that our paper was significantly improved during revisions.

1. In the Tv20S-MZB dataset (Fig. S10), only 2% of particles were fully assembled proteasomes while the remaining were half proteasomes. Please provide a full atlas of 2D classes in the first round which will better reflect the particle distribution of half and full proteasomes.

Please, find below a more comprehensive atlas of the original 2D classes from the first round of classification, which more accurately reflects the distribution of half and full proteasomes. The updated dataset included approximately 27,000 side-view particles. This additional data supports a clearer representation of the particle distribution and confirms the presence of both half and fully assembled proteasomes. The new 2D class averages are attached as Figure for Reviewer #1.

2. In the Tv20S-MZB dataset (Fig. S10), the particle number was reduced from 18,314 to 14,257 but the resolution was decreased from 2.79 to 2.86. But the authors state (inconsistently) in Method “cryo-EM imaging” section: The resolution was improved by 3D classification and removal of poor 3D classes and further reiterations including 2D classifications”.

We apologize for the confusion regarding the resolution changes and the particle numbers in the Tv20S-MZB dataset (Fig. S10). We have included an updated explanation in the figure legend to clarify this issue.

The resolution values stated—2.79 Å and 2.86 Å—represent different stages in the analysis process. The 2.79 Å resolution corresponds to the interim classification obtained from the 3D homogeneous refinement subroutine. The 2.86 Å resolution is the final resolution estimate based on the Fourier Shell Correlation (FSC).

This number is both affected by the subroutine (program that provides that data) and different number of particles after careful removal of poor-quality particles during the classification process to improve the dataset quality, this led to a slight decrease in the overall resolution number. This discrepancy is a result of removal of low-quality particles, which, while improving the dataset's integrity, slightly affected the final resolution.

We now state in the paper (Materials and Methods section): "The resolution changed from 2.79 Å to 2.86 Å during 3D classification."

3. In Method “cryo-EM imaging” section, particles and exposures were reduced to 6,135 particles by manual curation” – did the author mean micrographs rather than particles??

Yes, we apologize, we meant micrographs. This error has been corrected. And we now state in the paper: “.. particles and exposures were reduced to 6,135 micrographs by manual curation...”

4. In the validation report of PDB ID 8P0T, the particle number was 197,322, the electron dose was 50 e/Å², and the defocus was 800-3600 nm. However, the particle number in Fig. S10 is different. The dose was 40 e/Å² and the defocus was 800-2400 nm in Table S3.

We apologize for this inconsistency. We are thankful to Expert Reviewer #1 for spotting this unfortunate overlook. The errors originated from inputting setup used during grid screening using the Glacios microscope and these errors were, unfortunately, not corrected during the finalization. Both 8OIX and 8P0T final datasets were collected at the same KRIOS instrument with same and identical instrument setup with a was 40 e-/Å², and using magnification 165000x and defocus range (-2.4)–(-0.9) μm with the increments of 0.3 μm. The table S3 contains correct values including the final number of particles. We have reestablished communication with the database and asked for correction of experimental procedures within each deposition.

Figure S10 was corrected, these numbers were deleted.

5. In the validation report of PDB ID 8OIX, the electron dose was 50 e/Å², and the defocus was 800-3600 nm. However, the dose was 40 e/Å² and the defocus was 800-2400 nm in Table S3.

Same as above. We apologize for this inconsistency and are thankful to Expert Reviewer #1 for spotting this unfortunate overlook. Both 8OIX and 8P0T datasets were collected at the same KRIOS instrument with same and identical instrument setup with a was 40 e-/Å², and using magnification 165000x and defocus range (-2.4)–(-0.9) μm with the increments of 0.3 μm. We have reestablished communication with the database and asked for correction of experimental procedures within each deposition.

Table S3 now contains correct values including the final number of particles.

6. In the validation report of PDB ID 8OIX, the Unmasked-calculated resolution is 3.73 Å with 0.143 threshold and 6.91 Å with the 0.5 threshold. This is an unusually large gap, indicative of low signal-to-noise ratio, likely due to limited number of particle (14,257) in the final reconstruction. The author should either note this shortcoming or collect more data to address the problem.

We agree with the concern of Expert Reviewer #1 regarding the unusually large gap between the unmasked-calculated resolution (3.73 Å with 0.143 threshold and 6.91 Å with the 0.5 threshold) in the validation report of PDB ID 8OIX. This difference can indeed suggest a lower signal-to-noise ratio, potentially due to the limited number of particles (14,257) in the final reconstruction.

However, we have performed extensive iterations from scratch to avoid map bias. The purpose of applying a mask was to exclude data that are irrelevant to the project (half proteasomes in this case). After applying the mask, the resolution achieved was sufficient, as demonstrated by the quality of the real maps.

We believe that attempting to lower the gap between the unmasked and masked data resolutions would not contribute meaningful improvements to our study, as it would involve including redundant data. Moreover, our 8P0T map exhibits adequate similarity and accuracy compared to our 8OIX model, confirming the validity of our results. The 8OIX map, submitted earlier to the RCSB database, supports this consistency.

We now state in the manuscript (Materials and Methods section): " We have observed rather a large gap between the unmasked-calculated resolution (3.73 Å with 0.143 threshold and 6.91 Å with the 0.5 threshold).

7. In Fig. S3c, Tv20S is more abundant in the -Ump1 samples, does not appear to agree with the description that "in the absence of Ump-1, the eluted fraction from the streptavidin column contained more half-proteasomes, indicating incomplete assembly".

Figure S3c shows that the ratio of half-proteasome to full-proteasome is higher in the absence of Ump-1. This figure does not compare the abundance of full-proteasome in the presence and absence of Ump-1. Although the -Ump-1 lanes might seem brighter, this full-proteasome is immature due to non-functional $\beta 5$ subunits (Figures S3b & S3d). Since the native PAGE gel used here cannot differentiate between the mature, functional full-proteasome (made with Ump-1) and the immature, non-functional full-proteasome (made without Ump-1), then comparing the band intensities is unreliable. Therefore, this figure focuses solely on the ratio of half-proteasome to full-proteasome.

For clarity, we now updated the manuscript with the sentence: "In the absence of Ump-1, the eluted fraction from the streptavidin column contained a higher ratio of more half-proteasomes to full-proteasomes, indicating incomplete assembly".

8. In Fig. S3b,d, why did the -Ump1 sample lose b5 binding to the fluorogenic probe? Does Ump-1 affect the b5 maturation?

Fig. S3b and S3d revealed that the full-proteasome generated in the absence of Ump-1 does not have a functioning $\beta 5$ subunit. The connection between $\beta 5$ and Ump-1 has been previously described for the yeast proteasome by Velez and colleagues (PMID: 38600323). In that study they show that Ump-1 directly interacts with the propeptide sequence of $\beta 5$ and properly orientates it between the $\beta 6$ and $\beta 7$ subunits. Therefore, the interaction of $\beta 5$ and Ump-1 is key for proteasome maturation in yeast and the same appears to be true for Tv20S.

We have now added the following sentence to the results section: “The connection between Ump-1 and $\beta 5$ activity are supported by previous studies of the yeast proteasome where Ump-1 was shown to directly interact with the propeptide sequence of $\beta 5$ and properly orientate it between the $\beta 6$ and $\beta 7$ subunits” and included the reference.

9. Fig. 1e and Fig S3b: There is no corresponding band in the silver stain gel compared with the fluorogenic probe binding gel. Are they from the same gel?

These images are from the same gel. Below (Figure for Reviewer 1), we have attached the silver stained and fluorescent gel images side-by-side. In addition, we have merged the two gels into one image. The $\beta 5$ subunit is not visible on the silver-stained gel but can be readily detected using the fluorescent probe. Since all 14 subunits (7α and 7β) are present at the same ratio on this gel, then the visible bands on the silver stained gel are likely to consist of more than one subunit that co-migrate on the gel.

10. Fig. 2c: the curves for the $\beta 1$ of nTv20S and rTv20S never reached V_{max} , indicating low affinity or inaccessible to the $\beta 1$ site. Consider another substrate?

The $\beta 1$ substrate for Tv20S was developed following extensive substrate profiling studies and it is the most efficient substrate available for this enzyme. We have reported those substrate profiling studies in Fajtova 2024 (PMID: 37066163). The goal of Fig. 2C was to show that the native and recombinant enzymes had the same activity. This comparison can be done with a substrate that has a low K_m (such as Ac-GWYL-amc ($\beta 5$) and Ac-FRSR-amc ($\beta 2$)) or with a substrate that has a higher K_m such as Ac-RYFD-amc ($\beta 1$).

11. From Fig 1c,f, the purified rTv20S was intact, so the proteasome was apparently destroyed by MZB, raising the question of the inhibitor entry into the active site. Several scenarios can be pictured, and different scenario may result in different structure and kinetics. The author should address if half proteasome binds the inhibitor.

We did not generate images of the apo-enzyme and therefore we cannot directly compare the ratio of half proteasome to full-proteasomes in MZB-treated and untreated images. It is possible the MZB destabilizes the full-proteasome yielding an increase in half-proteasomes. For our structural studies, we used a preparation of Tv20S that was co-expressed with Ump-1. However, we later discovered that incubation of this purified preparation at room temperature for 24 to 72 hours decreases the amount of half-proteasome present (see Fig S3c). In our future studies, we will examine the time-dependent assembly of Tv20S.

Based on published proteasome assembly studies, it is unlikely that MZB or other active site inhibitors bind to the half-proteasomes because the propeptides of $\beta 1$ $\beta 2$ and $\beta 5$ are not processed until the full proteasome assembly has occurred. Therefore, the catalytic sites of half-proteasomes are unlikely to be accessible by an inhibitor.

Evidence using other protease inhibitors asl suggest that MZB does not bind to half-proteasomes. For example, the activity-based probe, Me4BodipyFL-Ahx3Leu3VS, binds covalently to all three catalytic subunits. However, we do not see labelling of half-proteasomes in insect cell lysates that express Tv20S with a high abundance of half-proteasomes. This is demonstrated by Fig. S3a, Lane 3 where Tv20S is expressed without Ump-1 yet there is no labelling of a half-proteasome bands on this gel.

12. Fig. 4e-f and in page 9: there is no fig 4f. Ile35 mentioned in the text but not labeled in the figure.

Thank you for pointing out this error. Fig 4e-f was changed to Fig. 4d-e and Ile35 is now labeled.

Reviewer #2 (Remarks to the Author):

This revised manuscript about structural relationships between two established proteasome inhibitors and the *Trichomonas vaginalis* 20S proteasome addresses many of the concerns raised by me and the other reviewers. In my view, a main advance in this manuscript is the description of a recombinant method of producing Tv proteasome for development of therapeutically useful inhibitors. That said, despite its usefulness for Tv, analogous methodology for other versions of proteasomes is now becoming more common and is therefore less novel. One recent paper not cited here (in their response the authors state that they are not aware.....) includes Adolf et al Nat Struc Mol Biol (<https://doi.org/101038/s41594-024-01268-9>); maybe this paper crossed during preparation of the current manuscript. The promise that the differences between the current 20S Tv structure and human 20S will lead to Tv-specific inhibitors remains to be determined. The description of structural features of Tv 20S with known inhibitors is useful new information.

Thank you for these comments. The paper by Adolf et al Nat Struc Mol Biol was published in April while our manuscript has been under review. In fact, they have a sentence in their manuscript that states "Recombinant systems have enabled the study of mature proteasome complexes." and have cited our preprint on bioRxiv.

One issue that I continue to find confusing (also raised by Reviewer 1) is the low percentage of fully assembled recombinant protein in the presence of inhibitor used for determination of the structure. That seems at odds with the apparent homogeneous preparations of purified protein on native gels. Is the inhibitor promoting dissociation? Why not compare Tv 20S +/- inhibitors on native gels? Did I miss something? On a related issue, the data in Figure S3c/d seem odd. The vast majority of the samples appears to be mature 20S.

We have added a new native page gel that confirms that the mature 20S proteasome does not dissociate upon binding to the inhibitor (see Fig S3e). As outlined above in our responses to Reviewer 1, we found that incubation of the purified Tv20S+ Ump-1 at room temperature for up to 72 hours decreases the amount of half-proteasomes in the preparation. We assume that these half-proteasomes become full proteasomes. However, we discovered this conversion only after we had generated the structures of Tv20S with MZB and with CP-17. Therefore, we have a high amount of half-proteasomes in our cryoEM grids but many of the

gel-based studies have much lower amounts of half-proteasomes, since these enzyme preparations were incubated at room temperature for 72 hours.

Reviewer #3 (Remarks to the Author):

The authors have nicely rewritten the manuscript to highlight the new elements of their work and performed additional experimentation in response to reviewer comments. Prior to publication, the K_m and k_{cat} data should be reported with values and units, not just raw traces. While the overall conclusions are still somewhat modest, other than the demonstration that the proteasome can be purified from Sf9 cells, the rewritten manuscript does make a strong point of indicating that different inhibitors target different active sites in the proteasome. Therefore, I recommend publication of a suitably revised manuscript with K_m and k_{cat} values in tabulated format with error bars and number of repeats indicated.

Thank you for your comments. We have now included the K_m and k_{cat} values in Fig. 2c.

REVIEWERS' COMMENTS

Reviewer #1 (Remarks to the Author):

Our concerns have been addressed.

Reviewer #2 (Remarks to the Author):

The authors have responded to my previous concerns satisfactorily. I have no further concerns.

Reviewer #1 (Remarks to the Author)

Our concerns have been addressed.

We are happy to hear that.

Reviewer #2 (Remarks to the Author)

The authors have responded to my previous concerns satisfactorily. I have no further concerns.

We are happy to hear that.